# Penalizing Infeasible Actions and Reward Scaling in Reinforcement Learning with Offline Data

Jeonghye Kim[2,1†]  Yongjae Shin[2,1†]  Whiyoung Jung[1]  Sunghoon Hong[1]
Deunsol Yoon[1]  Youngchul Sung[2]  Kanghoon Lee[1]  Woohyung Lim[1]

## Abstract

Reinforcement learning with offline data suffers from Q-value extrapolation errors. To address this issue, we first demonstrate that linear extrapolation of the Q-function beyond the data range is particularly problematic. To mitigate this, we propose guiding the gradual decrease of Q-values outside the data range, which is achieved through reward scaling with layer normalization (RS-LN) and a penalization mechanism for infeasible actions (PA). By combining RS-LN and PA, we develop a new algorithm called PARS. We evaluate PARS across a range of tasks, demonstrating superior performance compared to state-of-the-art algorithms in both offline training and online fine-tuning on the D4RL benchmark, with notable success in the challenging AntMaze Ultra task.

## 1. Introduction

Reinforcement learning (RL) enables agents to develop optimal decision-making strategies through real-time interactions. However, these interactions with real-world environments can expose the agent to considerable risks. To mitigate these risks, Offline RL, which derives optimal policies from pre-collected data, has emerged as a critical area of research (Fujimoto & Gu, 2021; Tarasov et al., 2024). Additionally, agents trained with offline RL can be deployed in real-world environments to further acquire knowledge, leading to the development of offline-to-online RL approaches (Lee et al., 2022; Nakamoto et al., 2024; LEI et al., 2024). However, due to the limited coverage of offline data, these methods often suffer from extrapolation error, where the Q-values of out-of-distribution (OOD) actions are overesti-

[†]Work done during an internship at LG AI Research. [1]LG AI Research, Seoul, Republic of Korea [2]School of Electrical Engineering, KAIST, Daejeon, Republic of Korea. Correspondence to: Woohyung Lim <w.lim@lgresearch.ai>.

*Proceedings of the 42nd International Conference on Machine Learning*, Vancouver, Canada. PMLR 267, 2025. Copyright 2025 by the author(s).

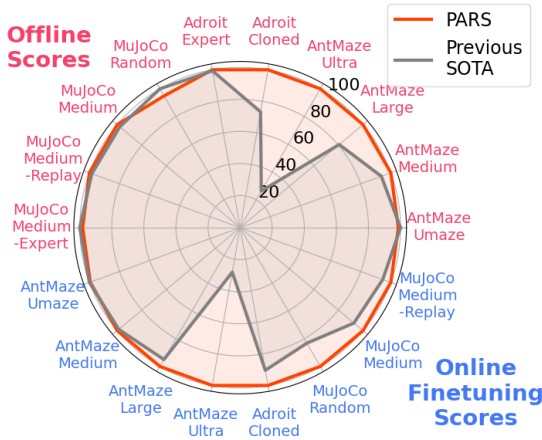

Figure 1: Comparison of PARS and prior SOTA, with scores normalized to each task's maximum performance.

mated, limiting the overall performance (Kumar et al., 2020; Kostrikov et al., 2022; Mao et al., 2024; Kim et al., 2024).

We first demonstrate that a key factor behind extrapolation error in RL with offline data is the tendency for linear extrapolation beyond the data range. ReLU-based MLPs often converge to linear functions outside the observed data (Agarap, 2018; Xu et al., 2021; Yue et al., 2024), leading to persistent boundary trends. When this trend is upward, it causes Q overestimation for OOD actions. To mitigate this, recent studies (Ball et al., 2023; Yue et al., 2024) propose applying layer normalization (LN; Ba et al., 2016) to the Q-function. Ball et al. (2023) shows that LN constrains Q-function predictions by bounding them to the norm of weight layers. However, while LN provides this bound, it does not sufficiently control Q-values outside the data range, leaving the problem unresolved without online interaction.

Therefore, RL with offline data requires methods that not only bound but also effectively reduce Q-values outside the data range. To address this, we propose two approaches: **(1) reward scaling with layer normalization (RS-LN)** and **(2) penalizing infeasible actions (PA)**. Our analysis shows that increasing the reward scale with LN reduces the perceived similarity by the function approximator between actions within the data range and those outside it. As a

result, gradient updates for in-distribution (ID) actions have a weaker influence on OOD Q-value predictions, leading to a reduction in OOD Q-values beyond the data range. In contrast to previous studies that adjust reward with their own manipulation methods (Kumar et al., 2020; Kostrikov et al., 2022; Tarasov et al., 2024), especially in AntMaze, we offer a novel perspective on reward scaling for OOD mitigation and demonstrate the effectiveness. Additionally, to further enforce a gradual decline in Q-values beyond the data range, we penalize the Q-values of infeasible actions far from the agent's feasible action regions.

At the end of the discussion, we introduce PARS (**P**enalizing infeasible **A**ctions and **R**eward **S**caling), built on the minimalistic TD3+BC framework (Fujimoto & Gu, 2021). PARS is simple to implement, requiring only a few extra lines of code. Without excessive conservatism or auxiliary models, PARS seamlessly transitions to online fine-tuning. Evaluated across diverse RL tasks, it consistently matches or surpasses prior state-of-the-art performance (Figure 1). Notably, PARS excels in the challenging AntMaze Ultra task, uniquely enabling successful offline-to-online training, demonstrating its robustness and effectiveness.

## 2. Preliminaries

The RL problem is formulated as a Markov Decision Process (MDP, Puterman, 1990) $\mathcal{M} = \langle \rho_0, \mathcal{S}, \mathcal{A}, P, \mathcal{R}, \gamma \rangle$, where $\rho_0$ is the initial state distribution, $\mathcal{S}$ is the state space, $\mathcal{A}$ is the action space, $P(s_{t+1}|s_t, a_t)$ is the transition probability, $\mathcal{R}(s_t, a_t)$ is the reward function, and $\gamma \in (0, 1)$ is the discount factor. In this study, we focus on scenarios with a continuous action space. We extend the action space, typically limited to feasible actions, by defining the infeasible action space $\mathcal{A}_I$ to account for potential extrapolation into infeasible regions by neural networks. Thus, the action space $\mathcal{A}$ is given by the union of the **feasible action region** $\mathcal{A}_F$, typically confined to a compact subset of $\mathbb{R}^n$, and the **infeasible action region** defined as $\mathcal{A}_I = \mathbb{R}^n \setminus \mathcal{A}_F$, which consists of actions the agent cannot perform in any state.

**Neural network extrapolation.** Xu et al. (2021) examine the behavior of neural networks when extrapolating, specifically focusing on what they learn beyond the scope of the training distribution. Their analysis reveals that ReLU MLPs struggle to effectively extrapolate in most nonlinear tasks due to their linear extrapolation. As predictions move further from the training data, they quickly transition to linear behavior along directions originating from the origin.

**Neural tangent kernel (NTK).** One factor that can influence network extrapolation is the extent to which a network update at one point affects the update at another. This can be measured using the neural tangent kernel (NTK) (Jacot et al., 2018), defined as $K_\phi(x, x') = \langle \nabla_\theta f_\theta(x), \nabla_\theta f_\theta(x') \rangle$.

The NTK determines how updates generalize across inputs in gradient-based learning. A high similarity in gradients implies that the nonlinear function $f_\theta$ perceives the two inputs $x$ and $x'$ as similar points (Yue et al., 2024).

**Reducing Q function extrapolation error with layer normalization.** Layer normalization (LN; Ba et al., 2016) stabilizes neural network training by normalizing the outputs of hidden layers. LN recenters and rescales these outputs using the transformation: for the feature vector of the $i$-th layer $\mathbf{h}_i$, $\hat{\mathbf{h}}_i = \frac{\mathbf{h}_i - \mu_i}{\sqrt{\sigma_i^2 + \epsilon}} \odot \boldsymbol{\eta}_i + \boldsymbol{\beta}_i$, for some $\epsilon > 0$, where $\odot$ denotes the elementwise product, $\mu_i$ and $\sigma_i^2$ are the mean and variance of the elements of $\mathbf{h}_i$, respectively, and $\boldsymbol{\eta}_i$ and $\boldsymbol{\beta}_i$ are learnable rescaling and shifting parameters, respectively. In recent RL studies, LN has been shown to enhance training stability, leading to improved final performance (Ball et al., 2023; Yue et al., 2024; Nauman et al., 2024). Ball et al. (2023) demonstrated that LN can constrain the Q-function prediction by bounding it to the norm of the weight layers. Moreover, Yue et al. (2024) provide a theoretical explanation for LN's effectiveness in mitigating Q-function divergence through NTK (Jacot et al., 2018) analysis.

## 3. Extrapolation Error in RL with Offline Data

### 3.1. Extrapolation Error for OOD Actions Outside the Convex Hull of In-distribution Samples

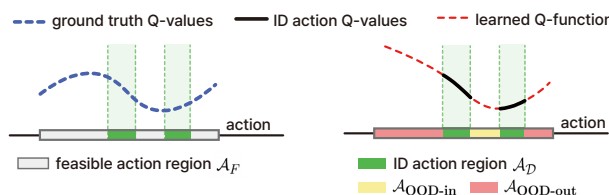

Figure 2: Composition of offline data and comparison between ground truth Q-values and learned Q-function

To gain deeper insight into extrapolation errors in RL with offline data, we revisit the nature of offline data $\mathcal{D}$. For a given state $s$, the associated action set $\mathcal{A}_s \subseteq \mathcal{A}_F$, can form multiple distinct clusters within the feasible action region $\mathcal{A}_F$, as shown in Figure 2. $\mathcal{A}_F$ is typically centered at the origin in many RL tasks, e.g., $\mathcal{A}_F = [-1, 1]^n$, where $n$ is the action dimension. In this scenario, OOD actions are defined as $\mathcal{A}_{\text{OOD}}(s) = \{a \in \mathcal{A}_F \mid a \notin \mathcal{A}_s\}$, and a convex hull can be constructed from $\mathcal{A}_s$ as follows:

$$\text{Conv}(\mathcal{A}_s) = \left\{ \sum_{i=1}^n \lambda_i a_i \; \middle| \; \lambda_i \geq 0, \sum_{i=1}^n \lambda_i = 1, a_i \in \mathcal{A}_s \right\}.$$

Subsequently, any OOD action $a \in \mathcal{A}_{\text{OOD}}(s)$ can be categorized as:

$$a \in \begin{cases} \mathcal{A}_{\text{OOD-in}}(s), & \text{if } a \in \text{Conv}(\mathcal{A}_s), \\ \mathcal{A}_{\text{OOD-out}}(s), & \text{if } a \notin \text{Conv}(\mathcal{A}_s). \end{cases}$$

As shown in Figure 2, extrapolation becomes significantly more challenging in $\mathcal{A}_{\text{OOD-out}}(s)$. As discussed in Xu et al. (2021); Yue et al. (2024), ReLU-based MLPs tend to behave linearly outside the data range, generalizing by extrapolating increasing or decreasing trends observed at the boundary of the convex hull. Since no training data exists for $\mathcal{A}_{\text{OOD-out}}(s)$, the Q-function in $\mathcal{A}_{\text{OOD-out}}(s)$ remains unconstrained, leading to uncontrollable error growth.

To investigate this, we analyzed how the max-Q action, which maximizes the learned Q-function for a given state, differs from the dataset actions across various D4RL (Fu et al., 2020) datasets, similar to the approach of Yue et al. (2024). We trained $Q_\phi$ using observed state-action pairs from the dataset (offline SARSA; Sutton, 2018; Kumar et al., 2022) and then derived the max-Q policy using the loss function $\mathcal{L}_{\text{max-Q}}(\phi) = \mathbb{E}_{s \sim \mathcal{D}}\left[\max_a Q_\phi(s, a)\right]$.

Table 1: Mean action norm for dataset actions and max-Q policy actions, with and without LN, normalized by the possible maximum norm, averaged over 5 seeds.

| Datasets | Data actions | Max-Q actions (w/o LN) | Max-Q actions (w/ LN) |
|---|---|---|---|
| hopper-medium | 0.62 | 0.95 | 0.94 |
| pen-cloned | 0.84 | 0.99 | 0.99 |
| antmaze-umaze-diverse | 0.81 | 0.99 | 0.99 |
| antmaze-ultra-diverse | 0.81 | 0.98 | 0.98 |

As shown in Table 1, the average action norm in the dataset is around 0.6-0.8, while actions generated by the max-Q policy almost always lie at the end of $\mathcal{A}_{\text{OOD-out}}(s)$ meeting the boundary of $\mathcal{A}_F$. This behavior persists even with LN, indicating that LN alone can fail to address the issue of linear extrapolation. This problem can become increasingly severe as the dimensionality of the action space $\mathcal{A}$ increases. If Q-values rise in $\mathcal{A}_{\text{OOD-out}}(s)$ along even a single action dimension, unintended actions may be selected.

### 3.2. Extrapolation for RL with Offline Data

*"What is the good extrapolation for RL with offline data?"*

A neural network cannot know the ground truth trends beyond the data range. Therefore, in offline RL, the Q-value in $\mathcal{A}_{\text{OOD-out}}(s)$ must be lower than the maximum Q-value within the data range to ensure the selection of optimal actions from within the data. Thus, extrapolation is required to ensure that the curve in $\mathcal{A}_{\text{OOD-out}}(s)$ remains below the in-data maximum, flattening or declining from the boundary of the convex hull $\text{Conv}(\mathcal{A}_s)$.

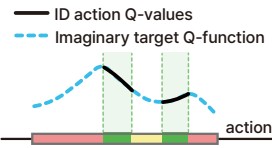

Figure 3: The target Q-function the Q-function needs to extrapolate to.

## 4. Penalizing Infeasible Actions and Reward Scaling

### 4.1. Reward Scaling Combined with LN (RS-LN)

Consider training a Q-function $Q_\phi$ with a positive TD target. Since the network is initialized with weights centered around zero, $Q_\phi$ initially produces small outputs, which gradually increase as it learns to match the targets. During training, the network generalizes across inputs, so learning from one input influences the outputs of other inputs deemed similar by $Q_\phi$ (Yue et al., 2024). If $Q_\phi$ considers actions in $\mathcal{A}_{\text{OOD-out}}(s)$ to be dissimilar from in-distribution actions $\mathcal{A}_\mathcal{D}$, then the gradient updates that push Q-values higher have a weaker effect on $\mathcal{A}_{\text{OOD-out}}(s)$. As a result, Q-values for $\mathcal{A}_{\text{OOD-out}}(s)$ do not increase as much, leading to their natural suppression relative to $\mathcal{A}_\mathcal{D}$.

Therefore, our goal is to encourage $Q_\phi$ to better distinguish between $\mathcal{A}_\mathcal{D}$ and $\mathcal{A}_{\text{OOD-out}}(s)$, which requires higher feature resolution in the learned representations. How can this be achieved simply? To build intuition, consider approximating $y = x$ over the interval $[0, 1]$ using a piecewise constant function with 5 equal-width bins. The maximum approximation error in this

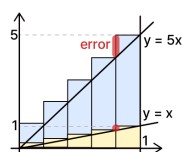

Figure 4: Error in approximating $y = x$ and $y = 5x$

case is 0.2. If we increase the scale of the function to $y = 5x$, the maximum error increases proportionally to 1. Reducing this error requires a finer partition of the input space, i.e., a higher resolution. In other words, larger output scales inherently demand finer resolution to maintain approximation fidelity.

A similar principle applies to neural networks. When the output scale increases, small differences in the input lead to larger differences in the output, encouraging the network to learn more fine-grained and expressive features, as we will see shortly. However, this mechanism does not always yield positive outcomes. It is effective primarily when combined with LN. Note that even if we change the function to $y = 5x$, using five bins still results in the same approximation error if the input domain is reduced to $[0, 0.2]$. Therefore, to necessitate higher resolution for $y = 5x$, the input volume must remain the same in both cases. LN facilitates this by consistently confining the input volume to the unit sphere.

#### 4.1.1. DIDACTIC EXAMPLE

We analyze how reward scaling and LN jointly mitigate OOD Q-value overestimation by adapting a regression approach from Ball et al. (2023). The true Q-function is defined as $y = f(x_1, x_2) = c_{\text{reward}} \cdot \left(\sqrt{x_1^2 + x_2^2}\right)$, where $c_{\text{reward}}$ is the reward scaling factor, and the feasible input region is $(x_1, x_2) \in [-1, 1]^2$. We generate a dataset

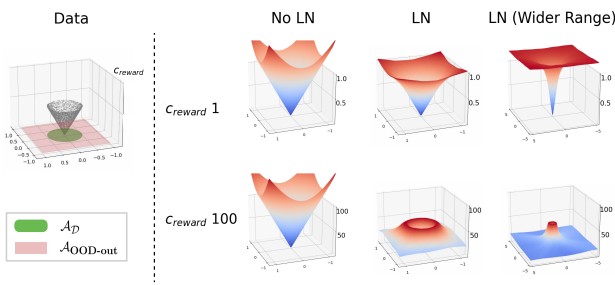

Figure 5: Results of training on a toy dataset using ReLU MLPs with vanilla regression (the 'No LN' column) and with LN (the 'LN' column), varying $c_{reward}$. The 'LN (Wider Range)' column shows results under the same conditions as 'LN', but with the $x_1$ and $x_2$ axes extended to the range $[-5, 5]$. For more examples, please refer to Appendix D.

$\{(x_1, x_2, y)\}$, where $y$ is defined only for inputs satisfying $x_1^2 + x_2^2 \leq 0.5^2$. Consequently, $\mathcal{A}_{\text{OOD-out}}(s)$ refers to the remaining region within $(x_1, x_2) \in [-1, 1]^2$ where $x_1^2 + x_2^2 > 0.5^2$. We fit the data using a 3-layer MLP (256 hidden units, ReLU activation) for both vanilla regression and regression with LN, varying $c_{reward}$ from 1 to 100.

As illustrated in the 'No LN' column, the Q-values exhibit catastrophic overestimation in $\mathcal{A}_{\text{OOD-out}}(s)$. Integrating LN helps mitigate the overestimation caused by linear extrapolation; however, $\mathcal{A}_{\text{OOD-out}}(s)$ remains overestimated relative to the in-distribution region. As the reward scale increases with LN applied (see the 'LN' column), the predictions in $\mathcal{A}_{\text{OOD-out}}(s)$ become noticeably lower. This effect becomes more apparent when the $x_1$ and $x_2$ axes are extended to the range $[-5, 5]$ (see the 'LN (Wider Range)' column). With LN and a higher reward scale, the Q-values in $\mathcal{A}_{\text{OOD-out}}(s)$ remain closer to zero overall, making them appear relatively lower than those in $\mathcal{A}_{\mathcal{D}}$. Note that reward scaling is effective when used together with LN, as expected.

To investigate this further, we plot the normalized NTK map to analyze the gradient similarity between the data boundary point $p = (0.32, -0.38)$ and the entire feasible action region $\mathcal{A}_F$. As shown in Figure 6, when $c_{reward} = 1$, high gradient similarity is observed between $p$ and $\mathcal{A}_{\text{OOD-out}}(s)$. In contrast, as $c_{reward}$ increases to 100, the similarity decreases, indicating that the network perceives the OOD data as less similar to the training data. Consequently, the influence of positive gradient updates from the ID region ($x_1^2 + x_2^2 \leq 0.5^2$) on the OOD region is reduced during training, naturally leading to downward extrapolation.

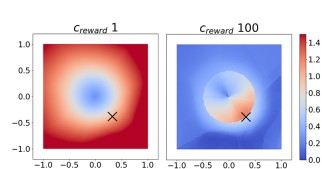

Figure 6: The normalized NTK map between $p = (0.32, -0.38)$ and $\mathcal{A}_F$ for $c_{reward} = 1, 100$.

### 4.1.2. RL EXAMPLE

We extend our analysis of RS-LN to real RL tasks. In various RL contexts, neural network expressivity has been studied especially in online RL (Kumar et al., 2021; Sokar et al., 2023; Kim et al., 2023; Obando Ceron et al., 2024). In particular, Sokar et al. (2023) introduced the concept of dormant neurons (neurons with zero activations) in RL, demonstrating that an increase in dormant neurons correlates with the network's underutilization and degraded performance.

Figure 7 shows the evolution of Q-value, dormant neuron ratio, and normalized return as $c_{reward}$ increases, with and without LN, when training TD3+BC on the AntMaze-medium-diverse dataset with $\gamma = 0.995$. More examples can be found in Appendix G. As shown in the figure, more ReLU neurons are activated up to 50 % to increase expressivity as $c_{reward}$ increases with LN, resulting in overall performance improvement. (Note that 100 % activation rate corresponds to linear operation and 50 % seems the maximum expressivity ratio for ReLU.) This result indeed aligns with our analysis that increasing $c_{reward}$ enhances feature resolution or expressivity. To the best of our knowledge, we are the first to show that network expressivity, as evaluated by dormant neurons, is linked to OOD mitigation in RL with offline data.

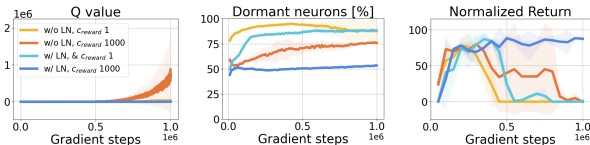

Figure 7: Plots illustrating how each metric changes during training in AntMaze-medium-diverse-v2 with TD3+BC.

### 4.2. Penalizing Infeasible Actions (PA)

While RS-LN aims not to increase the Q-values of $\mathcal{A}_{\text{OOD-out}}(s)$ to positive values from initial zero through the reduced NTK between $\mathcal{A}_{\mathcal{D}}$ and $\mathcal{A}_{\text{OOD-out}}(s)$, we additionally impose a hard constraint to guide the function's behavior at the convex hull boundary to trend downward. That is, we introduce penalties in regions far from the feasible region (infeasible action regions, $\mathcal{A}_I$), ensuring that the downward trend persists while minimizing its impact on Q-value predictions within $\mathcal{A}_F$.

To achieve this, in addition to the standard TD loss, we consider the PA loss to constrain the Q-value in $\mathcal{A}_I$ to $Q_{\min}$.

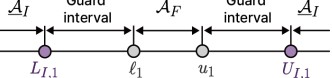

Figure 8: $\mathcal{A}_F$ and $\underline{\mathcal{A}}_I$ for $n = 1$

Since we want the Q-function in $\mathcal{A}_F$ to be sufficiently well estimated with the dataset and not heavily affected by the constraint in $\mathcal{A}_I$, we allow some guard interval between the feasible action region and the constraint-imposed $\mathcal{A}_I$.

Therefore, we consider the following subset of $\mathcal{A}_I$:

$$\underline{\mathcal{A}}_I = \bigcup_{i=1}^{n} \{(-\infty, L_{I,i}] \cup [U_{I,i}, \infty)\}, \tag{1}$$

where $n$ is the dimensions of action. Note that $\mathcal{A}_F$ is defined as $\bigcap_{i=1}^{n} \{(\ell_i, u_i)\}$, where $\ell_i$ and $u_i$ denote the lower and upper bounds of the feasible space in the $i$-th dimension. Thus, $L_{I,i} < \ell_i$ and $u_i < U_{I,i}$ to allow the guard or transition interval. Then, the PA loss is determined as

$$\mathcal{L}_{\text{PA}} = \min_{\phi} \mathbb{E}_{s \sim \mathcal{D}, a \in \underline{\mathcal{A}}_I} \left[ (Q_\phi(s,a) - Q_{\min})^2 \right], \tag{2}$$

where $Q_{\min}$ can be calculated as $c_{\text{reward}} \cdot r_{\min}/(1-\gamma)$. If the minimum reward of the task is unknown, it can be estimated from the dataset's minimum reward, as suggested in Mao et al. (2024). By combining two losses with a weighting factor $\alpha$, our modified TD loss is given as follows:

$$\mathcal{L}_{\text{Total}} = \min_{\phi} \left\{ \mathbb{E}_{s,a,s' \sim \mathcal{D}} \left[ (Q_\phi(s,a) - \mathcal{T}(s,a,s'))^2 \right] \right.$$
$$\left. + \alpha \cdot \mathbb{E}_{\substack{s,s' \sim \mathcal{D} \\ a \in \underline{\mathcal{A}}_I}} \left[ (Q_\phi(s,a) - Q_{\min})^2 \right] \right\}, \tag{3}$$

where $\mathcal{T}(s,a,s')$ is defined as $c_{\text{reward}} \cdot r(s,a) + \gamma \mathbb{E}_{a' \sim \pi_\theta(\cdot|s')} Q_\phi(s',a')$.

### 4.3. PARS Algorithm

Combining RS-LN and PA, we present a novel algorithm that prevents Q-value extrapolation error to ensure stable Q-learning across both offline and online fine-tuning phases: **P**enalizing infeasible **a**ctions and **r**eward **s**caling (PARS). PARS is based on the minimalist offline RL algorithm TD3+BC, and an extension to the in-sample learning methods (Kostrikov et al., 2022; Xu et al., 2023; Garg et al., 2023; Hansen-Estruch et al., 2023) can be found in Section 6.5. In addition, we provide a theoretical analysis of PARS in Appendix C.

**Infeasible action sampling.** The expectation over $a \in \underline{\mathcal{A}}_I$ in eq. (2) is practically realized by sample expectation or sample mean. To sample actions from the support in eq. (1), we consider the following uniform distribution for infeasible action sampling:

$$a_i \sim \begin{cases} \text{Uniform}(L_{I,i} - \Delta_L, \ L_{I,i}) & \text{for } a_i \leq L_{I,i}, \\ \text{Uniform}(U_{I,i}, \ U_{I,i} + \Delta_U) & \text{for } a_i \geq U_{I,i}, \end{cases} \tag{4}$$

As we shall see in Section 6.4, the performance does not heavily depend on the values of $L_{I,i}, U_{I,i}$ when these values are set as 100 to 1000 times the boundary of $\mathcal{A}_F$.

**Critic ensemble.** Following prior works (Ghasemipour et al., 2022; Lee et al., 2022; Ball et al., 2023), PARS can

also utilize a critic ensemble with a limited number of critics (4 for AntMaze, 10 for MuJoCo, and Adroit). Appendix G discusses its impact. For policy evaluation, we adopt the same approach as in Ball et al. (2023). For policy improvement, we use the minimum critic value during offline training, while averaging a subset of critics during online fine-tuning to avoid restricting online exploration.

## 5. Related Work

**Critic regularization in offline RL** To address the extrapolation error in offline RL, various critic regularization methods have been proposed. An et al. (2021) showed that increasing the number of critics in an ensemble can provide effective critic regularization. However, relying solely on ensembles for sufficient regularization can require up to 500 ensembles depending on the dataset, and even then, challenging tasks such as AntMaze may not be effectively solved (Tarasov et al., 2022; 2024).

Additionally, Kumar et al. (2020); Lyu et al. (2022); Mao et al. (2024) introduce a penalty to reduce Q-values for OOD actions that deviate from the behavior policy $\mu$. Unlike our approach, which focuses more on $\mathcal{A}_{\text{OOD-out}}(s)$ and addresses critic regularization from a more Q-network-centric perspective, prior works do not distinguish between $\mathcal{A}_{\text{OOD-out}}(s)$ and $\mathcal{A}_{\text{OOD-in}}(s)$, instead regulating them together by utilizing density differences or an auxiliary behavior model to better differentiate OOD actions within the data support. However, these approximation-based methods face increased uncertainty as the action dimension grows or tasks become more complex. In addition, penalizing OOD actions in $\mathcal{A}_{\text{OOD-in}}(s)$ can also impact Q-value predictions for nearby in-distribution actions. Moreover, online fine-tuning introduces the challenge of adapting density differences and auxiliary models learned offline to the online setting.

**Offline-to-online RL methods.** The policy trained on offline can be fine-tuned with additional online interaction, but this often causes distributional shifts between offline and online data (Nair et al., 2020; Lee et al., 2022; Uchendu et al., 2023). To tackle this, various approaches have been proposed based on existing offline algorithms (Lee et al., 2022; Nakamoto et al., 2024; Zhang et al., 2023a; Beeson & Montana, 2022). However, traditional offline algorithms are already designed with limited datasets in mind, leading to a conservative learning approach that can limit performance in online fine-tuning. To overcome this, Uni-O4 (LEI et al., 2024) proposes removing conservatism during the offline phase to facilitate a smoother transition to the online phase. Recently, Zhao et al. (2024) proposed ENOTO, effectively utilizing ensembles for efficient offline-to-online RL. Compared to existing methods, we do not propose new specialized techniques for online fine-tuning, enabling smooth transitions and superior performance with PARS.

Table 2: Offline PARS evaluation on the AntMaze, Adroit, and MuJoCo domains. We report the final normalized score averaged over five random seeds, with ± representing the standard deviation.

| AntMaze | TD3+BC | IQL | CQL | SAC-N | EDAC | MCQ | MSG | SPOT | SAC-RND | ReBRAC | PARS |
|---|---|---|---|---|---|---|---|---|---|---|---|
| antmaze-u | 78.6 | 87.5 | 74.0 | 0.0 | 0.0 | 27.5 | **97.9** | 93.5 | **97.0** | 97.8 | 93.8±2.1 |
| antmaze-u-d | 71.4 | 62.2 | 84.0 | 0.0 | 0.0 | 0.0 | 79.3 | 40.7 | 66.0 | 88.3 | **89.9**±7.5 |
| antmaze-m-p | 10.6 | 71.2 | 61.2 | 0.0 | 0.0 | 0.0 | 85.9 | 74.7 | 38.5 | 84.0 | **91.2**±3.9 |
| antmaze-m-d | 3.0 | 70.0 | 53.7 | 0.0 | 0.0 | 0.0 | 84.6 | 79.1 | 74.7 | 76.3 | **92.0**±2.2 |
| antmaze-l-p | 0.2 | 39.6 | 15.8 | 0.0 | 0.0 | 0.0 | 64.3 | 35.3 | 43.9 | 60.4 | **84.8**±5.9 |
| antmaze-l-d | 0.0 | 47.5 | 14.9 | 0.0 | 0.0 | 0.0 | 71.2 | 36.3 | 45.7 | 54.4 | **83.2**±5.6 |
| antmaze-ultra-p | 0.0 | 13.3 | 16.1 | 0.0 | 0.0 | 0.0 | 0.6 | 4.4 | 20.6 | 22.4 | **66.4**±4.4 |
| antmaze-ultra-d | 0.0 | 14.2 | 6.5 | 0.0 | 0.0 | 0.0 | 1.0 | 12.0 | 10.5 | 0.8 | **51.4**±11.6 |
| average | 20.5 | 50.7 | 40.8 | 0.0 | 0.0 | 3.4 | 60.6 | 47.0 | 49.6 | 60.6 | **81.6** |
| **Adroit** | **TD3+BC** | **IQL** | **CQL** | **SAC-N** | **EDAC** | **MCQ** | **SVR** | **SPOT** | **SAC-RND** | **ReBRAC** | **PARS** |
| pen-cloned | 61.5 | 77.2 | 39.2 | 64.1 | 68.2 | 35.3 | 65.6 | 15.2 | 2.5 | 91.8 | **107.5**±15.8 |
| pen-expert | 146.0 | 133.6 | 107.0 | 87.1 | -1.5 | 121.2 | 119.9 | 117.3 | 45.4 | **154.1** | 152.7±1.0 |
| door-cloned | 0.1 | 0.8 | 0.4 | -0.3 | **9.6** | 0.2 | 1.1 | 0.0 | 0.2 | 1.1 | 4.3±6.1 |
| door-expert | 84.6 | 105.3 | 101.5 | -0.3 | 106.3 | 73.0 | 83.3 | 0.2 | 73.6 | 104.6 | **106.0**±0.2 |
| hammer-cloned | 0.8 | 1.1 | 2.1 | 0.2 | 0.3 | 5.2 | 0.5 | 2.5 | 0.1 | 6.7 | **23.3**±20.8 |
| hammer-expert | 117.0 | 129.6 | 86.7 | 25.1 | 28.5 | 75.9 | 103.3 | 86.6 | 24.8 | **133.8** | 133.5±0.4 |
| relocate-cloned | -0.1 | 0.2 | -0.1 | 0.0 | 0.0 | -0.1 | 0.0 | -0.1 | 0.0 | 0.9 | **1.2**±0.7 |
| relocate-expert | 107.3 | 106.5 | 95.0 | -0.3 | 71.9 | 82.5 | 59.3 | 0.0 | 3.4 | 106.6 | **110.5**±1.5 |
| average | 64.7 | 69.3 | 54.0 | 22.0 | 35.4 | 49.2 | 54.1 | 27.8 | 18.8 | 75.0 | **79.9** |
| **MuJoCo** | **TD3+BC** | **IQL** | **CQL** | **SAC-N** | **EDAC** | **MCQ** | **SVR** | **SPOT** | **SAC-RND** | **ReBRAC** | **PARS** |
| halfcheetah-r | 11.0 | 13.1 | 17.5 | 28.0 | 28.4 | 28.5 | 27.2 | 23.8 | 29.0 | 29.5 | **30.4**±0.7 |
| hopper-r | 8.5 | 7.9 | 7.9 | **31.3** | 25.3 | **31.8** | **31.0** | 31.2 | 31.3 | 8.1 | 25.4±11.5 |
| walker2d-r | 1.6 | 5.4 | 5.1 | **21.7** | 16.6 | 17.0 | 2.2 | 5.3 | 21.5 | 18.4 | **21.8**±0.2 |
| halfcheetah-m | 48.3 | 47.4 | 44.0 | **67.5** | 65.9 | 64.3 | 60.5 | 58.4 | **66.6** | 65.6 | 64.2±1.2 |
| hopper-m | 59.3 | 66.3 | 58.5 | 100.3 | 101.6 | 78.4 | **103.5** | 86.0 | 97.8 | 102.0 | **104.1**±0.4 |
| walker2d-m | 83.7 | 78.3 | 72.5 | 87.9 | 92.5 | 91.0 | 92.4 | 86.4 | 91.6 | 82.5 | **97.3**±2.5 |
| halfcheetah-m-r | 44.6 | 44.2 | 45.5 | **63.9** | 61.3 | 56.8 | 52.5 | 52.2 | 54.9 | 51.0 | 57.0±0.6 |
| hopper-m-r | 60.9 | 94.7 | 95.0 | 101.8 | 101.0 | 101.6 | **103.7** | 100.2 | 100.5 | 98.1 | **103.1**±0.6 |
| walker2d-m-r | 81.8 | 73.9 | 77.2 | 78.7 | 87.1 | 91.3 | **95.6** | 91.6 | 88.7 | 77.3 | 95.8±1.4 |
| halfcheetah-m-e | 90.7 | 86.7 | 91.6 | **107.1** | 106.3 | 87.5 | 94.2 | 86.9 | **107.6** | 101.1 | 103.0±2.4 |
| hopper-m-e | 98.0 | 91.5 | 105.4 | 110.1 | 110.7 | 111.2 | 111.2 | 99.3 | 109.8 | 107.0 | **113.1**±0.3 |
| walker2d-m-e | 110.1 | 109.6 | 108.8 | **116.7** | 114.7 | 114.2 | 109.3 | 112.0 | 105.0 | 111.6 | 111.8±0.7 |
| average | 58.2 | 59.9 | 60.8 | 76.3 | 76.0 | 72.8 | 73.6 | 69.4 | 75.4 | 71.0 | **77.3** |

# 6. Experiments

## 6.1. Experiment Setup

**Benchmark.** We use three domains (AntMaze, Adroit, and MuJoCo) with a total of 28 datasets from the D4RL benchmark (Fu et al., 2020). In the performance comparison table, we used the following abbreviations: u for umaze, m for medium, l for large, p for play, d for diverse, r for random, m-r for medium replay, and e for expert. For a detailed explanation of the benchmark please refer to Appendix F.1.

**Hyperparameters.** We uniformly set $L_{I,i} = L_I$ and $U_{I,i} = U_I$ across all action dimensions $i$. We use $\Delta_L = |L_I|$ and $\Delta_U = |U_I|$, where $L_I$ and $U_I$ are scaled by either 100 or 1000 times the boundary of $\mathcal{A}_F$. Given that in the tasks considered, $L_I < 0$ and $U_I > 0$, we sample infeasible actions from the intervals $[2L_I, L_I]$ and $[U_I, 2U_I]$, respectively. Furthermore, we enable the tuning of around 10 sets of hyperparameters, including $\alpha$ and TD3+BC's $\beta$, similar to prior works (Wu et al., 2022; Nikulin et al., 2023; Tarasov et al., 2024). For AntMaze, strong performance can be achieved even with a single hyperparameter set, as

shown in Appendix G. For detailed hyperparameters and implementation, please refer to Appendix F.3.

## 6.2. Baseline Comparison

**Offline training.** We evaluate PARS in comparison with 10 prior SOTA baselines: TD3+BC (Fujimoto & Gu, 2021), IQL (Kostrikov et al., 2022), CQL (Kumar et al., 2020), SAC-N (An et al., 2021), EDAC (An et al., 2021), MSG (Ghasemipour et al., 2022), SPOT (Wu et al., 2022), SVR (Mao et al., 2024), SAC-RND (Nikulin et al., 2023), and ReBRAC (Tarasov et al., 2024). The details of the baselines are described in Appendix F.2. We primarily used official scores from the respective papers. If a score was unavailable for certain dataset versions, we referenced other benchmarking papers or conducted our own experiments, tuning the algorithm with the recommended hyperparameters.

The evaluation results are summarized in Table 2. As shown, while other algorithms often perform well in specific domains but falter in others, PARS consistently demonstrates robust performance across a diverse range of domains. For

Table 3: PARS evaluation on the AntMaze, Adroit, and MuJoCo domains after fine-tuning with 300k online samples. We report the final normalized score averaged over five random seeds, with ± indicating the standard deviation. The corresponding performance graphs are in Appendix A.

| Antmaze | CQL[1] | SPOT | PEX | RLPD | Cal-QL | ReBRAC | PARS |
|---|---|---|---|---|---|---|---|
| antmaze-u | **99.0**±0.6 | 98.4±1.9 | 95.2±1.6 | **99.4**±0.8 | 90.1±10.8 | **99.4**±0.9 | **99.7**±0.8 |
| antmaze-u-d | 76.9±39.7 | 55.2±32.8 | 34.8±30.1 | **99.2**±1.0 | 75.2±35.0 | 97.4±2.1 | 97.8±2.1 |
| antmaze-m-p | 94.4±3.0 | 91.2±3.8 | 83.4±2.3 | 97.4±1.4 | 95.1±6.3 | 96.8±1.9 | **99.1**±1.8 |
| antmaze-m-d | 98.8±2.5 | 91.6±3.5 | 86.6±5.0 | 98.6±1.4 | 96.3±4.8 | 95.8±3.6 | **99.4**±1.1 |
| antmaze-l-p | 87.3±5.6 | 60.4±21.5 | 56.0±3.9 | 93.0±2.5 | 75.0±14.7 | 71.4±30.9 | **96.2**±3.0 |
| antmaze-l-d | 65.3±28.3 | 69.4±23.7 | 60.4±6.8 | 90.4±3.9 | 74.4±11.8 | 89.0±3.4 | **96.8**±1.2 |
| antmaze-ultra-p | 21.3±19.0 | 0.0±0.0 | 13.3±5.8 | 8.8±16.5 | 6.9±2.7 | 0.0±0.0 | **86.5**±4.4 |
| antmaze-ultra-d | 6.3±6.6 | 5.8±11.5 | 26.7±11.3 | 40.0±37.0 | 5.7±11.2 | 1.0±1.7 | **86.4**±6.5 |
| average | 68.7 | 59.0 | 57.1 | 78.4 | 64.8 | 68.9 | **95.2** |
| **Adroit** | CQL | Off2On | SPOT | RLPD | Cal-QL | ReBRAC | PARS |
| pen-cloned | -2.6±0.1 | 102.5±166.0 | 117.1±13.4 | **154.8**±11.8 | -1.6±1.6 | 134.1±7.2 | **155.4**±3.1 |
| door-cloned | -0.34±0.00 | -8.0±0.2 | 0.05±0.06 | **110.8**±6.1 | -0.34±0.0 | 53.3±35.3 | 102.1±26.8 |
| hammer-cloned | 0.24±0.03 | -7.4±0.4 | 90.2±23.2 | 139.7±5.6 | 0.21±0.08 | 114.4±10.3 | **141.5**±1.9 |
| relocate-cloned | -0.33±0.01 | -1.5±0.5 | -0.29±0.04 | 4.8±7.1 | -0.34±0.01 | 1.5±1.1 | **53.8**±7.7 |
| average | -0.8 | 21.4 | 51.8 | 102.5 | -0.5 | 75.8 | **113.2** |
| **MuJoCo** | CQL | Off2On | PEX | RLPD | Cal-QL | Uni-O4 | PARS |
| halfcheetah-r | 26.5±3.4 | 92.7±5.7 | 60.9±5.0 | 91.5±2.5 | 32.9±8.1 | 6.8±3.9 | **100.1**±2.9 |
| hopper-r | 10.0±1.5 | 95.3±9.2 | 48.5±38.9 | 90.2±19.1 | 17.7±26.0 | 12.4±1.8 | **109.7**±5.3 |
| walker2d-r | 12.4±7.9 | 27.9±2.2 | 9.8±1.6 | 87.7±14.1 | 9.4±5.6 | 5.7±0.8 | **113.9**±13.9 |
| halfcheetah-m | 78.9±1.3 | 103.3±1.4 | 70.4±2.3 | 95.5±1.5 | 77.0±2.2 | 56.6±0.8 | **107.0**±5.0 |
| hopper-m | 100.9±0.6 | 106.3±1.7 | 86.2±26.3 | 91.4±27.8 | 100.7±0.8 | 104.8±2.6 | **111.5**±0.4 |
| walker2d-m | 88.7±0.4 | 109.8±29.6 | 91.4±14.3 | 121.6±2.3 | 97.0±8.2 | 106.5±3.4 | **126.4**±2.1 |
| halfcheetah-m-r | 50.3±28.3 | 95.6±1.7 | 55.4±5.1 | 90.1±1.3 | 62.1±1.1 | 53.2±5.4 | **98.5**±1.0 |
| hopper-m-r | 103.9±1.8 | 101.7±14.8 | 95.3±7.2 | 78.9±24.5 | 101.4±2.1 | 103.4±6.6 | **107.0**±1.4 |
| walker2d-m-r | 105.4±1.8 | 120.3±9.4 | 87.2±13.6 | 119.0±2.1 | 98.4±3.3 | 115.5±2.9 | **130.1**±4.4 |
| average | 64.1 | 96.4 | 67.2 | 96.2 | 66.3 | 62.8 | **111.6** |

instance, SAC-N (An et al., 2021) shows strong performance in MuJoCo, but it struggles to learn effectively in AntMaze. In contrast, ReBRAC excels in Adroit but falls behind other algorithms in both MuJoCo and AntMaze. Specifically, in the challenging AntMaze Large and Ultra datasets, PARS achieves approximately 24% and 280% performance improvements over existing baselines, respectively. These impressive results, achieved without excessive computational resources (as shown in Appendix H) and through a straightforward implementation (as shown in Appendix E), have the potential to advance the practical application of offline RL.

**Online fine-tuning.** After offline training, we conduct online fine-tuning with 300K of online samples and compare its score with 8 prior SOTA baselines: CQL (Kumar et al., 2020), Off2On (Lee et al., 2022), SPOT (Wu et al., 2022), RLPD (Ball et al., 2023), PEX (Zhang et al., 2023a), Cal-QL (Nakamoto et al., 2024), Uni-O4 (LEI et al., 2024), ReBRAC (Tarasov et al., 2024). We reproduced the results using the official implementations for all baseline scores in online fine-tuning, with details provided in Appendix F.2.

The experimental results are presented in Table 3, and the corresponding performance graphs can be found in Ap-

pendix A. Observing the results, except for two datasets, PARS outperforms all baselines across all datasets. For the online phase, RLPD, which leverages an offline dataset while learning from scratch online without an explicit offline training phase, showed strong performance compared to other baselines. This aligns with previous findings suggesting that offline-trained policies and critics can sometimes hinder online fine-tuning (Zhang et al., 2023b; Kong et al., 2024; Zhang et al., 2024; Zhou et al., 2025). However, PARS surpassed RLPD, challenging these findings and highlighting that online fine-tuning can be a highly effective framework when proper critic regularization is applied. Notably, in the cases of AntMaze ultra-play, ultra-diverse, and Adroit relocate-cloned, PARS is the only algorithm to demonstrate strong performance in these challenging scenarios.

PARS enhances network expressivity by increasing the reward scale, enabling swift adaptation to novel online samples. Additionally, the progressive reduction of Q-values at $\mathcal{A}_{\text{OOD-out}}(s)$ facilitates online exploration by using offline data as an anchor for initiating exploration, thereby significantly narrowing the search space. This approach is particularly effective for online fine-tuning, especially in complex tasks like AntMaze Ultra, where online interaction steps are limited.

---

[1]CQL can be fine-tuned with SAC (Haarnoja et al., 2018), as proposed in Nakamoto et al. (2024).

### 6.3. Comparing PARS with Goal-Conditioned Offline RL in AntMaze

We further compare PARS with recent goal-conditioned offline RL baselines tailored for complex, long-horizon tasks like AntMaze (Kostrikov et al., 2022; Yang et al., 2022; Hejna et al., 2023; Park et al., 2024b; Zeng et al., 2024), which leverage techniques such as goal relabeling (Yang et al., 2022; Hejna et al., 2023), hierarchical frameworks (Park et al., 2024b), and advanced architectures like transformers (Zeng et al., 2024). Details are in Appendix F.2.

Table 4: Performance comparison of PARS on AntMaze with goal-conditioned offline RL baselines.

|  | GC-IQL | WGCSL | DWSL | HIQL | GCPC | PARS |
|---|---|---|---|---|---|---|
| u | 91.6 | 90.8 | 71.2 | 79.2 | 71.2 | **93.8**±2.1 |
| u-d | 88.8 | 55.6 | 74.6 | 86.2 | 71.2 | **89.9**±7.5 |
| m-p | 82.6 | 63.2 | 77.6 | 84.1 | 70.8 | **91.2**±3.9 |
| m-d | 76.2 | 46.0 | 74.8 | 86.8 | 72.2 | **92.0**±2.2 |
| l-p | 40.0 | 0.6 | 15.2 | **86.1** | 78.2 | 84.8±5.9 |
| l-d | 29.8 | 2.4 | 19.0 | **88.2** | 80.6 | 83.2±5.6 |
| ultra-p | 20.6 | 0.2 | 25.2 | 39.2 | 56.6 | **66.4**±4.4 |
| ultra-d | 28.4 | 0 | 25.0 | 52.9 | **54.6** | 51.4±11.6 |
| avg | 57.3 | 32.4 | 47.8 | 75.3 | 69.4 | **81.6** |

In Table 4, PARS surpasses goal-conditioned baselines not only on the challenging ultra dataset but also in the overall average score across the AntMaze domain. These results highlight that even in sparse, long-horizon tasks, fundamental off-policy RL with proper regularization can excel without specialized designs or architectures.

### 6.4. Discussion on the Components of PARS

**How does each component of PARS affect offline performance?**

To identify the source of PARS's performance gains, we evaluate offline performance across different settings: the critic network without LN or PA (None), with only PA (PA), with only LN (LN), and with both LN and PA (LN & PA), while varying the reward scale $c_{\text{reward}}$. The

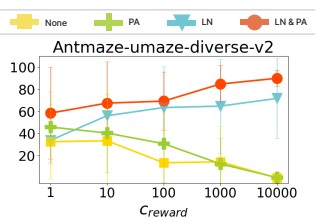

Figure 9: PARS offline performance, with varying $c_{\text{reward}}$ and the application of LN and PA.

results are shown in Figure 9, with error bars indicating the standard deviation. As illustrated, PA helps mitigate extrapolation error, although its effect diminishes at larger reward scales. The presence of LN leads to a consistent trend of improved performance as $c_{\text{reward}}$ increases. Furthermore, combining LN with PA results in even higher performance, aligning with the earlier discussion in Section 4.2. Additional ablation results are provided in Appendix G.

**How significant is the infeasible action penalty in online fine-tuning?** PA can be applied not only during offline training but also in the online fine-tuning phase. To investigate its benefits in this setting, we conducted additional experiments by varying the use of PA during fine-tuning, following offline training with PA applied. The results, shown in Figure 10, indicate that removing PA during online fine-tuning after using it in offline training led to instability. In contrast, maintaining PA throughout resulted in more stable learning and significant performance improvements.

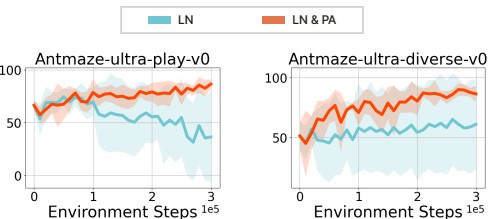

Figure 10: Ablation on PA: normalized score over 300k online fine-tuning steps.

**How far should infeasible actions be from the feasible action region?** As described in Section 4.3, we sample infeasible actions far from the feasible region. The distance is determined by $|L_I|$ and $|U_I|$, and for the benchmark comparison, we set $|L_I| = |U_I|$. Figure 11 demonstrates the impact of this configuration. When $|L_I|$ is small, meaning the sampled infeasible actions are near the feasible region, suboptimal performance is noted, indicating that penalizing infeasible actions may influence policy evaluation within the feasible region.

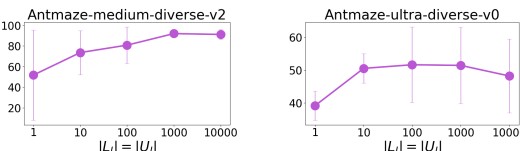

Figure 11: Final normalized score of PARS averaged over five random seeds with varying $|L_I| = |U_I|$.

### 6.5. Extension to In-Sample Learning-Based Methods

TD3+BC is an actor-critic method that may sample OOD actions during training. Alternatively, in-sample learning-based methods (Kostrikov et al., 2022; Xu et al., 2023; Garg et al., 2023; Hansen-Estruch et al., 2023) avoid OOD action sampling by implicitly learning the maximum Q-values using only in-sample data. Building on this, we investigated whether RS-LN and PA could also be applied to such methods. Specifically, we conducted experiments with IQL (Kostrikov et al., 2022), a representative in-sample learning approach, using a discount factor $\gamma = 0.995$ and

increasing the temperature parameter $\tau$ to 20. We also varied $\gamma$ and $\tau$ to report the best-performing results for IQL. For RS-LN, the reward scaling factor $c_{reward}$ was set to 1000.

Table 5: Performance improvement after applying RS-LN and PA to IQL. The scores are the averages of the final evaluations across five random seeds.

| AntMaze | IQL (reproduce) | IQL with RS-LN (Ours) | IQL with PA and RS-LN (Ours) |
|---|---|---|---|
| u-d | 66.5±5.5 | **83.0**±6.2 | 81.1±7.9 |
| l-p | 45.4±5.8 | **60.4**±6.2 | **60.6**±5.3 |
| ultra-p | 13.3±5.7 | **36.6**±13.3 | **37.3**±10.7 |

As shown in Table 5, PA was not particularly effective in this setting, likely due to its nature as a method that avoids OOD sampling. Conversely, RS-LN demonstrated noticeable effectiveness, possibly because a more expressive Q-function approximator provides better estimates even for in-sample Q-values. However, it did not surpass the performance of TD3+BC-based PARS.

**Why does the combination of TD3+BC with RS-LN and PA lead to better results?** To understand this more deeply, let us assume the existence of a highly expressive policy capable of accurately capturing the in-sample actions. Then, consider the following two major policy extraction methods in offline RL:

**(1) Weighted behavioral cloning (e.g., AWR, IQL)**

$$\max_{\pi} \mathbb{E}_{s,a\sim D}\left[e^{\alpha(Q(s,a)-V(s))}\log \pi(a \mid s)\right], \quad (5)$$

with $\alpha$ to control the (inverse) temperature.

**(2) Behavior-constrained policy gradient (e.g., DDPG+BC, TD3+BC, diffusion-QL)**

$$\max_{\pi} \mathbb{E}_{s\sim\mathcal{D},\, a'\sim\pi(\cdot\mid s)}\left[Q(s,a')\right]-w\cdot\mathbb{E}_{s\sim\mathcal{D}}\left[\mathcal{R}(\pi(\cdot\mid s),\pi_{\beta}(\cdot\mid s))\right], \quad (6)$$

where $\mathbb{E}_{s\sim\mathcal{D}}\left[\mathcal{R}(\pi(\cdot\mid s),\pi\beta(\cdot\mid s))\right]$ is a behavior regularization term. Thus, a more expressive policy can better approximate the target distribution, such as $\exp\left(\alpha(Q(s,\cdot)-V(s))\right)$ in Eq. (5), which in turn increases the value of the objective function.

Consider the example illustrated in Figure 12 (a), which contains two disjoint in-sample action regions separated by an intermediate $\mathcal{A}_{OOD\text{-}in}$ region. Suppose we apply IQL's in-sample policy learning using Equation (5), with a perfectly learned Q-function and a highly expressive policy $\pi_{\theta}$ initialized near zero. The resulting policy (Figure 12 (b)) will match the Q-function within the in-sample regions but assign near-zero probability elsewhere.

As a result, the policy does not generate OOD actions. While this behavior reduces the risk of selecting poor actions, it

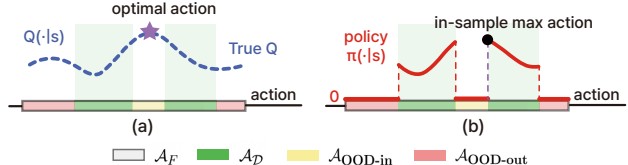

Figure 12: (a) True Q-values at a fixed state $s$ for the feasible action region $\mathcal{A}_{\mathcal{F}}$ (b) Learned policy obtained using a highly expressive policy.

also prevents the policy from discovering the globally optimal action when it lies within $\mathcal{A}_{OOD\text{-}in}$. Therefore, prior work (Park et al., 2024a) highlights that allowing a certain degree of deviation from the in-sample region, without straying too far, can be beneficial. Broadening the range of considered actions helps utilize the Q-function over a wider action space and enhances overall performance, assuming the Q-function is well trained. In addition, when considering online finetuning, assigning near-zero probability to all OOD actions can severely limit online exploration, restricting performance improvements.

To overcome the limitation of in-sample policy learning of IQL, we need to go beyond in-sample policy learning. One way is to consider the policy learning, eq. (6). e.g., TD3-BC in which $\mathcal{R}(\pi(\cdot\mid s),\pi_{\beta}(\cdot\mid s)) = |\pi(s) - a|^2$. Here, the policy is allowed to generate OOD action $a'$ from $s'$, but not too far from in-sample actions. Then, the Q-values of OOD actions, especially those in $\mathcal{A}_{OOD\text{-}out}$, matter now, and suppressing the upward trend in $\mathcal{A}_{OOD\text{-}out}$ becomes crucial. Here, RS-LN and PA play an important role, creating strong synergy with eq. (6), which leads to strong performance in both offline training and online finetuning.

## 7. Conclusion

We introduce PARS, aimed at preventing critic extrapolation error and enhancing overall performance in both offline training and online fine-tuning of RL with offline data. Our analysis of reward scaling with LN reveals that increasing the reward scale reduces the function approximator's perceived similarity between in-range and OOD actions, weakening the influence of gradient updates on OOD Q-values and leading to their reduction. Additionally, applying penalties to the infeasible action region can impose additional constraints to ensure that OOD Q-values beyond the data range trend downward. PARS demonstrated substantial performance improvements over previous SOTA across diverse RL tasks in both offline training and online fine-tuning phases. Our findings suggest a new perspective, departing from conventional views, that strong performance across a wide array of RL tasks is achievable with only simple adjustments to off-policy algorithms, provided that appropriate regularization for OOD mitigation is applied.

# Acknowledgements

This work was supported by LG AI Research and partly by the National Research Foundation of Korea(NRF) grant funded by the Korea government(MSIT) (No. RS-2025-00557589, Generative Model Based Efficient Reinforcement Learning Algorithms for Multi-modal Expansion in Generalized Environments).

# Impact Statement

This paper presents work whose goal is to advance the field of Machine Learning. There are many potential societal consequences of our work, none which we feel must be specifically highlighted here.

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

# A. Performance Graphs for Online Fine-Tuning (300K)

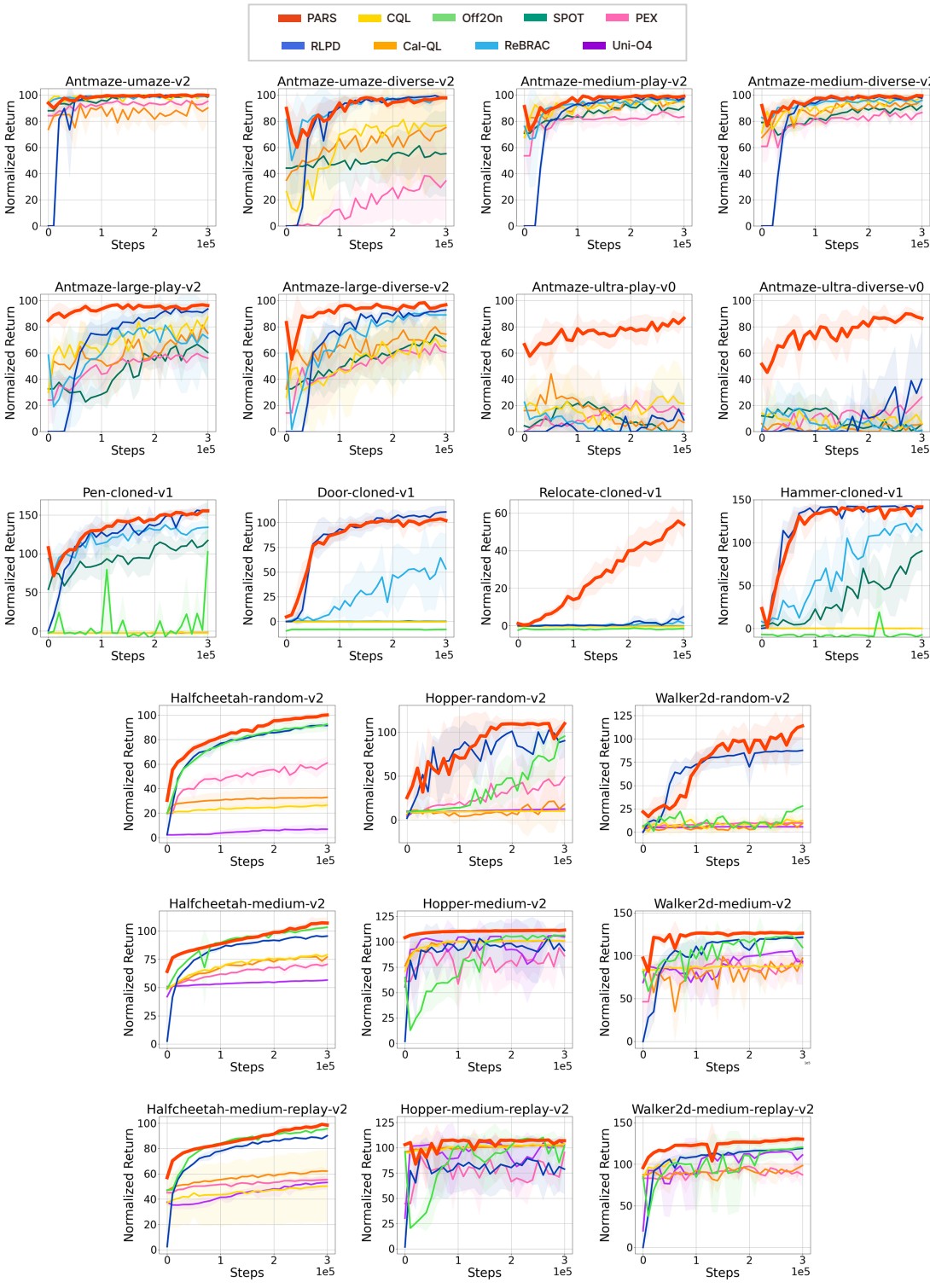

Figure 13: The performance graph of the online fine-tuning (300k), using five random seeds, corresponds to Table 3. The solid line indicates the mean, while the shaded region represents the standard deviation.

# B. Limitations and Future Research.

Although PARS has achieved superior performance, its approach is most effective when the target Q-value is positive. To naturally mitigate OOD, it is necessary to assume a positive Q-value so that reducing the impact of in-distribution data's gradient updates on OOD output leads to OOD value reduction. In RL, rewards can include both positive rewards and negative penalties. If the penalties outweigh the rewards, the Q-value can become negative. In such cases, the Q-function can be trained by adjusting the offline data rewards to ensure that the overall reward remains positive. Moreover, it still benefits from the support of a critic ensemble, particularly in MuJoCo and Adroit, where misestimated Q-values in OOD actions within the convex hull of the data can be further corrected by the ensemble. We expect that further investigation into data fitting challenges in limited datasets, coupled with potential refinements, could lead to robust performance with just a double-critic setup.

As a future research direction, exploring the most efficient way to apply LN to RL with offline data, both theoretically and empirically, would be valuable. Additionally, designing activation functions in combination with RS-LN or PA presents an intriguing research area.

# C. Theoretical Analysis

## C.1. Theoretical Justification of PARS

Here, we show the convergence of the PARS loss function defined in eq. (3). For this, we define the following sets:

$$
\begin{aligned}
\text{ID} &= \{(s, a) \mid \beta(a \mid s) > 0\} \\
\text{OOD}_{\text{in}} &= \text{ConvexHull}(\text{ID}) \setminus \text{ID} \\
\text{OOD}_{\text{out}} &= (\text{ConvexHull}(\text{ID}))^c
\end{aligned}
\tag{7}
$$

Now, we define the operator $T_{\text{pars}}^\pi$ as:

$$
T_{\text{pars}}^\pi Q(s, a) = \begin{cases}
T^\pi Q(s, a), & \text{if } (s, a) \in \text{ID}, \\
\mathbb{E}_{(s', a') \in \text{kNN}(s, a; \text{ID})}[T^\pi Q(s', a')], & \text{if } (s, a) \in \text{OOD}_{\text{in}}, \\
Q_{\min}, & \text{if } (s, a) \in \text{OOD}_{\text{out}},
\end{cases}
\tag{8}
$$

where $\text{kNN}(s, a; \text{ID})$ denotes the set of the k-nearest neighbors within ID. We show that $T_{\text{pars}}^\pi$ is a contraction under the $|\cdot|_\infty$ norm:

$$
\begin{aligned}
&\text{Case 1:} \quad \left| T_{\text{pars}}^\pi Q_1 - T_{\text{pars}}^\pi Q_2 \right| = |T^\pi Q_1 - T^\pi Q_2| \leq \gamma \|Q_1 - Q_2\|_\infty \\
&\text{Case 2:} \quad \left| T_{\text{pars}}^\pi Q_1 - T_{\text{pars}}^\pi Q_2 \right| = |\mathbb{E}_{\text{kNN}}[T^\pi Q_1] - \mathbb{E}_{\text{kNN}}[T^\pi Q_2]| \leq \gamma \|Q_1 - Q_2\|_\infty \\
&\text{Case 3:} \quad |Q_{\min} - Q_{\min}| = 0 \leq \gamma \|Q_1 - Q_2\|_\infty
\end{aligned}
\tag{9}
$$

Thus, $T_{\text{pars}}^\pi$ is a contraction, and the Q-function converges accordingly.

In Case 2, computing $\text{kNN}(s, a; \text{ID})$ is computational expensive. Since the Q-function is approximated by a neural network, we allow it to implicitly learn this behavior rather than explicitly incorporating it into the loss function. For the same reason, assigning $Q_{\min}$ can unintentionally affect nearby values. To prevent this, we introduce a "guard interval" that excludes actions just outside $\mathcal{A}_F$ from being evaluated as $Q_{\min}$. In practice, a guard interval of about 1000 (see Figure 11) is sufficient to avoid influencing Q-updates.

## C.2. Comparison of TD3+BC and IQL Critic Updates and the Effectiveness of RS-LN

Following the discussion of applying RS-LN to in-sample learning-based methods in Section 6.5, we compare the critic update of TD3+BC (Fujimoto & Gu, 2021), as used in PARS, with the critic updates of other in-sample learning-based methods. We focus on a representative method, IQL (Kostrikov et al., 2022), to analyze in depth whether RS-LN can be broadly applied.

First, the TD3+BC critic update loss function is defined as:

$$
\mathcal{L}_{\text{Q}} = \min_\phi \mathbb{E}_{s, a, s' \sim \mathcal{D}} \left[ \left( Q_\phi(s, a) - \left( c_{\text{reward}} \cdot r(s, a) + \gamma \mathbb{E}_{a' \sim \pi_\theta(\cdot | s')} Q_\phi(s', a') \right) \right)^2 \right].
\tag{10}
$$

Additionally, the critic update loss function of IQL is expressed as:

$$\mathcal{L}_Q = \min_\phi \mathbb{E}_{s,a,s'\sim\mathcal{D}} \left[ (Q_\phi(s,a) - (c_{\text{reward}} \cdot r(s,a) + \gamma V_\psi(s')))^2 \right],$$ (11)

with an additional value function loss that aligns $V_\psi(s)$ with $Q_\phi(s,a)$:

$$\mathcal{L}_V = \min_\psi \mathbb{E}_{s,a\sim\mathcal{D}} \left[ \mathcal{L}_\tau^2 (Q_\phi(s,a) - V_\psi(s)) \right],$$ (12)

where expectile regression is defined as $\mathcal{L}_\eta^2(u) = |\eta - \mathbb{1}(u < 0)|u^2$, with $\eta \in [0.5, 1)$, to formulate the asymmetrical loss function for the value network $V_\psi$.

Therefore, the critic updates in TD3+BC and IQL can be expressed in a unified form. The generalized critic loss is as follows:

$$\mathcal{L}_{\text{Critic}} = \min_\phi \mathbb{E}_{s,a,s'\sim\mathcal{D}} \left[ (Q_\phi(s,a) - Y(s,a,s'))^2 \right],$$ (13)

where the target $Y(s,a,s')$ is defined as:

$$Y(s,a,s') = c_{\text{reward}} \cdot r(s,a) + \gamma G(s').$$ (14)

The key difference lies in the definition of the next-state value estimate $G(s')$:

$$G(s') = \begin{cases} \mathbb{E}_{a'\sim\pi_\theta(\cdot|s')}Q_\phi(s',a'), & \text{(TD3+BC)} \\ V_\psi(s'), & \text{(IQL)} \end{cases}.$$ (15)

Although the target $Y(s,a,s')$ varies between TD3+BC and IQL and is dynamically adjusted throughout the learning process rather than remaining fixed, the core task of fitting $Q_\phi$ to $Y$ follows a structurally similar approach in both methods. Each method minimizes the squared error between $Q_\phi$ and the changing $Y$, providing a common framework for understanding their update mechanisms. In this context, both methods share the characteristic that an increase in $c_{\text{reward}}$ leads to an increase in the target value $Y$ and that the application of LN helps mitigate catastrophic overestimation. As analyzed in Figures 5 and 7, this can have the same effect on regression fitting and network expressivity, suggesting the potential for the general applicability of RS-LN.

## D. More Discussion on Didactic Example

**Analyzing split data distributions.**  Figure 5 illustrate a scenario in which data is concentrated in a single region. Additionally, we examined a new toy dataset consisting of two slightly separated inverted cones and analyzed its learning characteristics.

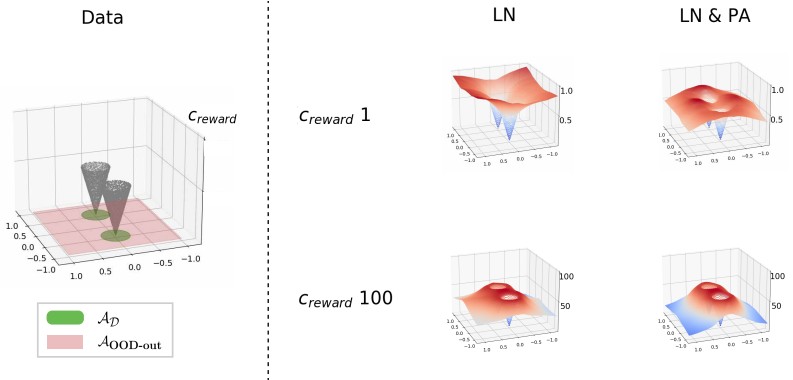

Figure 14: Results of fitting the two inverted, cone-shaped input datasets using an MLP network, with and without LN and PA, at a $c_{\text{reward}}$ of 1 or 100.

Figure 14 demonstrates the data fitting behavior when $c_{\text{reward}}$ is set to 1 or 100, comparing cases where only LN is applied (the 'LN' column) and where both LN and PA are applied (the 'LN & PA' column). As observed, LN mitigates OOD Q-value overestimation, and when the reward scale increases, the predicted values in $\mathcal{A}_{\text{OOD-out}}(s)$ even decrease relative to the in-distribution predictions, similar to the results in Figure 5. Additionally, when both LN and PA are applied, the Q-value in $\mathcal{A}_{\text{OOD-out}}(s)$ is further pushed downward, gradually decreasing as it moves further from the data range. Furthermore, LN enables smooth interpolation between the two split data regions.

**Impact of activation functions.** In Section 4.1, we demonstrated that RS-LN's effectiveness is tied to the network expressivity, which can be also influenced by the activation function (Raghu et al., 2017). The activation function plays a crucial role in determining whether neurons are activated or remain inactive during training. Accordingly, we examined how fitting characteristics, including OOD mitigation, vary depending on the activation function.

We tested the toy dataset from Figure 5 in a setting with $c_{\text{reward}} = 100$ and LN applied, using GELU (Hendrycks & Gimpel, 2016), Sigmoid, SiLU (Elfwing et al., 2018), as well as cases with no activation function. The results can be seen in Figure 15. While the Sigmoid prevents OOD Q-values from increasing, it does not reduce them as effectively as other activation functions. GELU reduces OOD Q-values, but it was overly conservative. Exploring the effect of activation functions on network expressivity could also be an interesting topic for future research. In the absence of activation, OOD Q-values remained high, even with a large $c_{\text{reward}}$. This effect extends beyond the didactic setting, influencing real-world RL performance, as detailed in Appendix G.

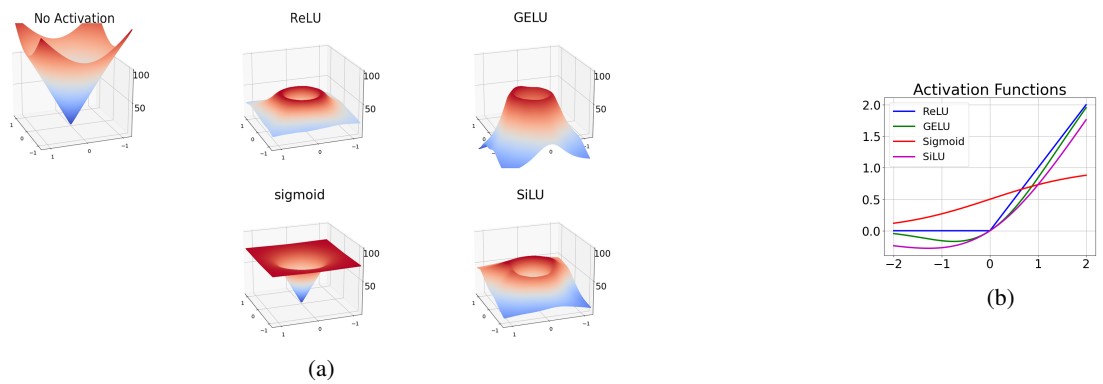

(a)

(b)

Figure 15: (a) The fitting results of applying various activation functions to the toy dataset from Figure 5 in a setting with $c_{\text{reward}} = 100$ and LN applied, (b) Plot of various activation functions.

# E. Reference Implementation

We provide a JAX-based reference implementation of the critic loss, and the complete training code is available at https://github.com/LGAI-Research/pars.

```
def _critic_loss(self,
      critic_params,
      critic_target_params,
      actor_target_params,
      transition,
      rng):
    state, action, next_state, reward, not_done = transition

    # Scale the reward by the predefined reward scaling factor
    reward = self.reward_scale * reward

    # Compute the target Q-value (implementation details omitted)
    target_Q = ...  # Placeholder for target Q-value computation

    # Get current Q estimates (implementation details omitted)
    current_Q = ...  # Placeholder for current Q estimate computation

    # Compute critic loss
    Q_loss = jnp.mean(jnp.square(current_Q - target_Q[:, 0][None, ...]))

    # Uniform infeasible action sampling
    infeasible_action = (jax.random.uniform(rng, action.shape) * 2) - 1
```

```
infeasible_action = jnp.where(infeasible_action < 0, infeasible_action - 1, infeasible_action + 1) * self.L

# Compute current Q-value for infeasible action
current_Q_infeasible = self.critic_model.apply(critic_params, state, infeasible_action)

# Set target Q-value for infeasible action to min_q
target_Q_infeasible = jnp.ones_like(target_Q) * self.min_q

# Compute infeasible Q loss
Q_loss_infeasible = jnp.mean(jnp.square(current_Q_infeasible - target_Q_infeasible[:, 0][None, ...]))

# Combine the losses
critic_loss = Q_loss + self.alpha * Q_loss_infeasible

return critic_loss
```

Listing 1: An example of reward scaling and infeasible action sampling implementation in JAX.

# F. Experiment Details

## F.1. Benchmark Details

Fu et al., 2020 introduced a variety of datasets designed for different RL tasks, such as AntMaze, Adroit, and MuJoCo. Additionally, zhengyao jiang et al., 2023 expanded the AntMaze domain by proposing an ultra dataset, featuring a larger map size than previously proposed, thus increasing the complexity of the task.

For the AntMaze domain, we leverage eight datasets: {umaze-v2, umaze-diverse-v2, medium-play-v2, medium-diverse-v2, large-play-v2, large-diverse-v2, ultra-play-v0, ultra-diverse-v0}. These datasets encompass different levels of difficulty based on the maze's size, complexity, and the diversity of start and goal positions.

In the Adroit domain, we utilize four tasks: {pen, door, hammer, relocate}, each associated with two dataset qualities: {cloned-v1, expert-v1}. The cloned datasets are created by training an imitation policy using demonstration data, executing the policy, and mixing the resulting data with the original demonstrations in a 50-50 ratio. In contrast, expert datasets are derived from fine-tuned RL policy.

For the MuJoCo domain, we use three tasks: {halfcheetah, hopper, walker2d}, each of which has four dataset qualities: {random-v2, medium-replay-v2, medium-v2, medium-expert-v2}. The random dataset consists of data generated by randomly initialized policies, the medium dataset comes from partially trained policies, the medium-replay dataset includes data from replay buffers during training, and the expert dataset contains demonstrations from well-trained agents performing near-optimal behavior.

## F.2. Baselines

**Offline training.** We primarily relied on the officially reported scores from each paper for datasets benchmarked for comparison. For datasets that were not benchmarked in the original paper, we referred to scores from other works that reported results for those datasets. For datasets without available scores from other sources, we reproduced the results using the respective implementations, tuning hyperparameters according to the recommendations in each paper.

Specifically, for datasets not benchmarked in the original paper, we either obtained the scores or conducted experiments as outlined below. In all other cases, we used the scores reported in the original paper. In cases where we conducted the experiments, we reported the final evaluation scores using five random seeds, with the mean and standard deviation provided in Tables 6 and 7.

- AntMaze: To begin, since the AntMaze Ultra datasets had not been benchmarked in previous studies we compared, we conducted experiments for all prior baselines. For MSG and SAC-RND, we ran the experiments using the official implementations. For TD3+BC, IQL, CQL, SPOT, and ReBRAC, we used the CORL library (Tarasov et al., 2022), which provides a single-file implementation of state-of-the-art (SOTA) offline RL algorithms. For the remaining datasets, we referenced the SAC-N and EDAC scores from Tarasov et al. (2022). Additionally, we sourced the MSG and SAC-RND scores from Tarasov et al. (2024), as those works benchmarked the v0 and v1 versions of the AntMaze datasets, rather than the v2 versions.

- Adroit: For Adroit, we obtained the TD3+BC, IQL, CQL, and SAC-RND scores from Tarasov et al. (2024), and the

SAC-RND and EDAC scores for expert datasets from Tarasov et al. (2022). Moreover, for SPOT and SVR, we ran the experiments using the CORL library and the official SVR implementation, respectively.

- MuJoCo: For MuJoCo, we obtained the IQL and CQL scores for random datasets from Lyu et al. (2022). Moreover, we ran SPOT on random datasets using the CORL library.

The URLs for each implementation are listed below:

- CORL - `https://github.com/tinkoff-ai/CORL`
- MSG - `https://github.com/google-research/google-research/tree/master/jrl`
- SAC-RND - `https://github.com/tinkoff-ai/sac-rnd`
- SVR - `https://github.com/MAOYIXIU/SVR`

Table 6: Final normalized evaluation scores averaged over five random seeds for SPOT and SVR. For each dataset, we tuned them as recommended in the paper and reported the best scores.

| Dataset | SPOT |
|---|---|
| halfcheetah-r | 23.8±0.5 |
| hopper-r | 31.2±0.4 |
| walker2d-r | 5.3±9.3 |
| pen-cloned | 15.2±18.7 |
| pen-expert | 117.3±14.9 |
| door-cloned | 0.0±0.0 |
| door-expert | 0.2±0.0 |
| hammer-cloned | 2.5±3.2 |
| hammer-expert | 86.6±46.3 |
| relocate-cloned | -0.1±0.0 |
| relocate-expert | 0.0±0.0 |

| Dataset | SVR |
|---|---|
| pen-cloned | 65.6±18.8 |
| pen-expert | 119.9±11.2 |
| door-cloned | 1.1±1.6 |
| door-expert | 83.3±14.9 |
| hammer-cloned | 0.5±0.4 |
| hammer-expert | 103.3±16.2 |
| relocate-cloned | 0.0±0.0 |
| relocate-expert | 59.3±10.2 |

| Dataset | MCQ |
|---|---|
| antmaze-u | 27.5±20.6 |
| antmaze-u-d | 0.0±0.0 |
| antmaze-m-p | 0.0±0.0 |
| antmaze-m-d | 0.0±0.0 |
| antmaze-l-p | 0.0±0.0 |
| antmaze-l-d | 0.0±0.0 |
| pen-cloned | 35.3±28.1 |
| pen-expert | 121.2±15.9 |
| door-cloned | 0.2±0.5 |
| door-expert | 73.0±2.2 |
| hammer-cloned | 5.2±6.3 |
| hammer-expert | 75.9±30.2 |
| relocate-cloned | -0.1±0.0 |
| relocate-expert | 82.5±7.2 |

Table 7: Final normalized evaluation scores averaged over five random seeds for the baselines on the AntMaze Ultra datasets. For each dataset, we tuned them as recommended in the paper and reported the best scores.

| Dataset | TD3+BC | IQL | CQL | MCQ | MSG | SPOT | SAC-RND | ReBRAC |
|---|---|---|---|---|---|---|---|---|
| antmaze-ultra-p | 0.0±0.0 | 13.3±5.7 | 16.1±8.5 | 0.0±0.0 | 0.6±0.9 | 4.4±1.3 | 20.6±15.0 | 22.4±11.7 |
| antmaze-ultra-d | 0.0±0.0 | 14.2±6.2 | 6.5±3.5 | 0.0±0.0 | 1.0±1.4 | 12.0±4.4 | 10.5±8.8 | 0.8±1.8 |

**Online fine-tuning.** During the online fine-tuning phase, since the official score for 300k online samples is typically unavailable, we re-run all the baselines using their corresponding official implementations, except for CQL, which uses the provided code from Cal-QL. The URLs for each implementation are listed below:

- RLPD - `https://github.com/ikostrikov/rlpd`
- Cal-QL - `https://github.com/nakamotoo/Cal-QL`
- Off2On - `https://github.com/shlee94/Off2OnRL`
- PEX - `https://github.com/Haichao-Zhang/PEX`
- Uni-O4 - `https://github.com/Lei-Kun/Uni-O4`
- SPOT - `https://github.com/thuml/SPOT`

**Goal-conditioned offline RL.** We obtain the HIQL scores from Park et al., 2024b, and the GC-IQL, WGCSL, DWSL, and GCPC scores from Zeng et al., 2024. For the umaze and umaze-diverse scores for HIQL, we run the experiments using the implementations provided for each respective algorithm in the official HIQL repository (`https://github.com/seohongpark/HIQL`). The resulting scores are as follows: umaze: 79.2±4.2, umaze-diverse: 86.2±5.7.

### F.3. PARS

We built our code on the JAX (Bradbury et al., 2018) implementation of TD3 (Fujimoto et al., 2018) (`https://github.com/yifan12wu/td3-jax`) and made modifications to suit the PARS algorithm, such as adding an infeasible action penalty, and reward scaling. For critic ensembles, we referenced the implementation of SAC-N (`https://github.com/Howuhh/sac-n-jax`).

Table 8: $Q_{\min}$ for each task.

| AntMaze | $Q_{\min}$ |
|---|---|
| antmaze-umaze | 0 |
| antmaze-medium | 0 |
| antmaze-large | 0 |
| antmaze-ultra | 0 |
| **Adroit** | $Q_{\min}$ |
| pen | -715 |
| door | -42 |
| hammer | -348 |
| relocate | 0 |
| **MuJoCo** | $Q_{\min}$ |
| halfcheetah | -366 |
| hopper | -166 |
| walker2d | -229 |

Table 9: PARS's general hyperparameters.

| Offline Hyperparameters | Value |
|---|---|
| optimizer | Adam (Kingma & Ba, 2015) |
| batch size | 256 |
| learning rate (all networks) | 3e-4 |
| tau ($\tau$) | 5e-3 |
| hidden dim (all networks) | 256 |
| gamma ($\gamma$) | 0.995 on AntMaze, 0.99 on others |
| infeasible region distance | 1000 on AntMaze, 100 on others |
| actor cosine scheduling | True on Adroit, False on others |
| nonlinearity | ReLU (Agarap, 2018) |
| **Online Fine-Tuning Hyperparameters** | **Value** |
| exploration noise | 0.1 on MuJoCo 0.05 on others |
| learning starts | 0 |
| update to data (UTD) ratio | 20 |

**AntMaze.** For offline training, we tuned $c_{\text{reward}}$ to 100, 1000, and 10,000, $\beta$ between 0.005 and 0.01, and $\alpha$ between 0.001 and 0.01. During online fine-tuning, we used a 50/50 mix of offline and online data. The online $\beta$ was tuned to 0, 0.001, and 0.01, while $\alpha$ was fixed at 0.001, except for ultra-play, which used 0.0001, and $S_{k_{\text{actor}}}$ was set to 1. The hyperparameters for each dataset are listed in Table 10.

**Adroit.** For offline training, we set $c_{\text{reward}}$ to 10 and tuned $\beta$ between 0.1 and 0.01, and $\alpha$ to 0.001 and 0.01. During online fine-tuning, a 50/50 mix of offline and online data was used. The online $\alpha$ was set to 0.001, and $\beta$ was tuned to 0 and 0.01, and $S_{k_{\text{actor}}}$ was set to 1. The specific hyperparameters for each dataset are provided in Table 10.

Table 10: Dataset-specific hyperparameters of PARS for AntMaze and Adroit domains used in offline training and online fine-tuning.

| | Offline | | | Online |
|---|---|---|---|---|
| **AntMaze** | $c_{\text{reward}}$ | $\beta$ | $\alpha$ | $\beta$ |
| antmaze-umaze | 10000 | 0.005 | 0.001 | 0 |
| antmaze-umaze-diverse | 10000 | 0.005 | 0.001 | 0.001 |
| antmaze-medium-play | 1000 | 0.01 | 0.001 | 0 |
| antmaze-medium-diverse | 1000 | 0.01 | 0.001 | 0 |
| antmaze-large-play | 1000 | 0.01 | 0.001 | 0.01 |
| antmaze-large-diverse | 10000 | 0.01 | 0.01 | 0.01 |
| antmaze-ultra-play | 100 | 0.01 | 0.001 | 0.001 |
| antmaze-ultra-diverse | 10000 | 0.01 | 0.01 | 0.01 |

| | Offline | | Online |
|---|---|---|---|
| **Adroit** | $\beta$ | $\alpha$ | $\beta$ |
| pen-cloned | 0.01 | 0.01 | 0 |
| door-cloned | 0.01 | 0.01 | 0.01 |
| hammer-cloned | 0.1 | 0.001 | 0 |
| relocate-cloned | 0.01 | 0.01 | 0.01 |
| pen-expert | 0.01 | 0.01 | - |
| door-expert | 0.1 | 0.001 | - |
| hammer-expert | 0.01 | 0.001 | - |
| relocate-expert | 0.1 | 0.001 | - |

**MuJoCo.** For offline training, we used a $c_{\text{reward}}$ of 5 for HalfCheetah and a $c_{\text{reward}}$ of 10 for Walker2d and Hopper. The $\beta$ was set to 0, and $\alpha$ was tuned between 0.01, 0.001 and 0.0001. Additionally, for this domain, we found that adjusting policy noise and $S_{k_{\text{critic}}}$ provides further benefits, so we varied the policy noise between 0 and 0.2, and $S_{k_{\text{critic}}}$ between 2 and 10. During online fine-tuning, we used 5% offline data for the HalfCheetah and random datasets, and 50% for the remaining datasets. We also tuned $\alpha$ among 0.1, 0.01, and 0.0001, and the critic sample size for policy improvement between 1 and 10. The specific hyperparameters for each dataset can be found in Table 11.

Table 11: Dataset-specific hyperparameters of PARS for MuJoCo domain used in offline training and online fine-tuning.

| | | Offline | | | Online | |
|---|---|---|---|---|---|---|
| **MuJoCo** | $\alpha$ | policy noise | $\mathcal{S}_{k_{\text{critic}}}$ | $\alpha$ | $\mathcal{S}_{k_{\text{actor}}}$ |
| halfcheetah-random | 0.0001 | 0.2 | 2 | 0.0001 | 1 |
| halfcheetah-medium | 0.0001 | 0 | 2 | 0.0001 | 10 |
| halfcheetah-medium-replay | 0.0001 | 0 | 2 | 0.0001 | 10 |
| halfcheetah-medium-expert | 0.0001 | 0.2 | 10 | - | - |
| hopper-random | 0.01 | 0.2 | 2 | 0.01 | 1 |
| hopper-medium | 0.01 | 0 | 10 | 0.1 | 1 |
| hopper-medium-replay | 0.01 | 0 | 10 | 0.1 | 1 |
| hopper-medium-expert | 0.0001 | 0.2 | 10 | - | - |
| walker2d-random | 0.01 | 0 | 10 | 0.0001 | 10 |
| walker2d-medium | 0.01 | 0 | 10 | 0.1 | 1 |
| walker2d-medium-replay | 0.01 | 0 | 10 | 0.01 | 1 |
| walker2d-medium-expert | 0.0001 | 0.2 | 10 | - | - |

# G. More Ablation Study

**Extended results on the impact of PARS components on offline performance.**    In addition to Figure 9 in Section 6.4, we conducted additional experiments on various datasets to further explore the impact of the PARS components in offline training. As shown in Figure 16, beyond AntMaze, the application of RS-LN in MuJoCo and Adroit leads to a general improvement in performance. Furthermore, incorporating PA results in a more robust enhancement.

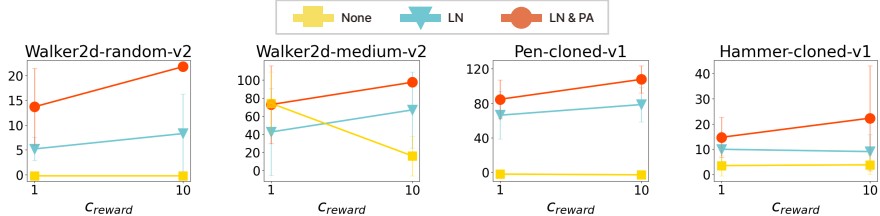

Figure 16: Extended ablation results of PARS components beyond Figure 9. We evaluate PARS with varying $c_{\text{reward}}$ and the application of LN and PA, averaged over five random seeds. The error bars represent the standard deviation.

**Impact of $c_{\text{reward}}$ on the dormant neuron ratio.**    In Section 4.1.2, we demonstrated that RS-LN effectively enhances network expressivity in the RL example, contributing to performance improvements. Additionally, we analyzed how the dormant neuron ratio changes with $c_{\text{reward}}$ when LN is applied across various datasets. Similarly to Section 4.1.2, we examined the dormant neuron ratio for $Q_\phi$, trained using TD3+BC with $\gamma = 0.995$.

Table 12: Dormant neuron ratio across various AntMaze datasets based on $c_{\text{reward}}$ while applying LN to TD3+BC $Q_\phi$.

| **AntMaze** | Dormant neurons [%] ($c_{\text{reward}} = 1$) | Dormant neurons [%] ($c_{\text{reward}} = 100$) |
|---|---|---|
| antmaze-umaze-diverse | 91±3.0 | 58±2.1 |
| antmaze-ultra-play | 93±2.1 | 57±1.5 |
| antmaze-ultra-diverse | 94±1.8 | 54±2.3 |

As shown in Table 12, the trend observed in Figure 7 remains consistent across different datasets. We confirmed that as the reward scale increases, the number of dormant neurons decreases, facilitating the utilization of higher feature resolution and enhancing network expressivity.

**AntMaze performance using a single hyperparameter setting.**    We further verified that comparable performance in the AntMaze domain can be achieved with just a single hyperparameter setting. This is particularly significant because many offline algorithms have struggled in the AntMaze domain, especially in the Ultra environment. Achieving significantly better performance than the baseline with a single hyperparameter setting in AntMaze, even though it is slightly lower than the PARS score presented in the paper, highlights a key advantage of PARS. The hyperparameters used in the single-hyperparameter experiment are a reward scale of 10,000, an alpha of 0.001, and a beta of 0.01.

Table 13: PARS AntMaze performance using a single hyperparameter setting. We averaged the scores over five random seeds, with $\pm$ indicating the standard deviation.

| AntMaze | other SOTA | other SOTA (goal-conditioned) | PARS | PARS (single hyperparam) |
|---|---|---|---|---|
| umaze | 97.9 (MSG) | 91.6 (GC-IQL) | 93.8 | 93.3$\pm$3.2 |
| umaze-diverse | 88.3 (ReBRAC) | 88.8 (GC-IQL) | 89.9 | 84.7$\pm$5.7 |
| medium-play | 85.9 (MSG) | 84.1 (HIQL) | 91.2 | 89.4$\pm$2.9 |
| medium-diverse | 84.6 (MSG) | 86.8 (HIQL) | 92.0 | 86.2$\pm$5.6 |
| large-play | 64.3 (MSG) | 86.1 (HIQL) | 84.8 | 77.5$\pm$1.8 |
| large-diverse | 71.2 (MSG) | 88.2 (HIQL) | 83.2 | 83.5$\pm$3.6 |
| ultra-play | 22.4 (ReBRAC) | 56.6 (GCPC) | 66.4 | 60.2$\pm$7.6 |
| ultra-diverse | 14.2 (IQL) | 54.6 (GCPC) | 51.4 | 50.8$\pm$11.1 |

**Sensitivity analysis of $\alpha$ hyperparameter.** When applying PA, the hyperparameter $\alpha$ in eq. (3) is important. It should be set such that assigning a $Q_{\min}$ penalty to infeasible actions does not interfere with the TD update for actions in the dataset. Accordingly, we set $\alpha$ to a small value in the range of 0.0001 to 0.001.

Table 14: PARS performance comparison across different values of $\alpha$. We averaged the scores over five random seeds, with $\pm$ indicating the standard deviation.

| AntMaze | $\alpha = 0.0001$ | $\alpha = 0.001$ | $\alpha = 0.01$ | $\alpha = 0.1$ | $\alpha = 1$ |
|---|---|---|---|---|---|
| umaze-diverse | 71.6$\pm$10.5 | **89.9**$\pm$7.5 | 88.6$\pm$5.9 | 87.5$\pm$6.9 | 82.3$\pm$8.2 |
| medium-diverse | 72.5$\pm$14.4 | **92.0**$\pm$2.2 | 91.7$\pm$3.5 | 91.2$\pm$2.8 | 40.5$\pm$25.4 |
| large-diverse | 79.5$\pm$8.2 | 83.0$\pm$4.9 | **83.2**$\pm$5.6 | 82.4$\pm$2.8 | 76.1$\pm$10.3 |
| ultra-diverse | 47.5$\pm$15.2 | 49.5$\pm$13.7 | **51.4**$\pm$11.6 | 50.9$\pm$9.5 | 45.4$\pm$13.1 |

We additionally conducted a sensitivity analysis on $\alpha$. As shown in Table 14, PARS performs well when $\alpha$ is in the range of 0.001 to 0.1. On the other hand, if $\alpha$ is too small (e.g., 0.0001), the penalty does not take effect properly, and if it is too large, it starts to interfere with the TD update on the dataset, leading to a decrease in performance.

**Impact of the number of critic ensembles.** As discussed in Section 4.3, PARS can be used in combination with a critic ensemble. We analyzed the effect of the number of critic ensembles on offline performance in Figure 17. As shown in the figure, while the impact is minimal in the Antmaze domain, incorporating an ensemble in the MuJoCo and Adroit domains enables more stable learning and thus contributes to improved performance.

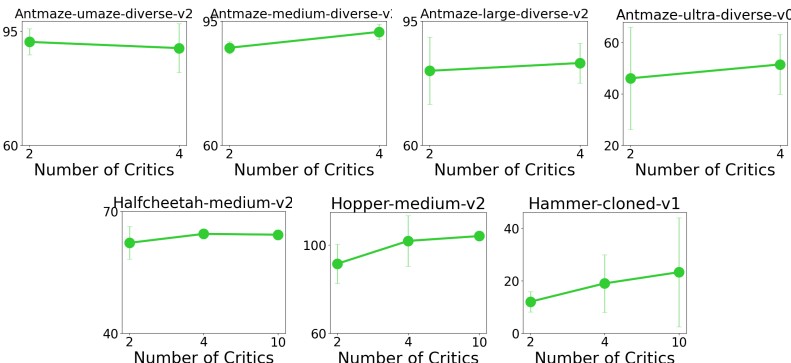

Figure 17: Final normalized score of PARS for offline training, averaged over five random seeds with varying numbers of critics. Error bars represent the standard deviation.

**Which activation function would be most compatible with PARS?** In Appendix D, we examine the effect of activation functions on fitting toy data. Additionally, we analyze how activation functions, when combined with PARS, influence performance in real RL tasks.

Figure 18 presents the results of applying each activation function to PARS, showing that the ReLU activation function consistently performs well across various tasks. While other activation functions outperform ReLU in some tasks, they lack robustness across all tasks. The impact of activation functions on RL tasks, in conjunction with RS-LN and PA, could be an

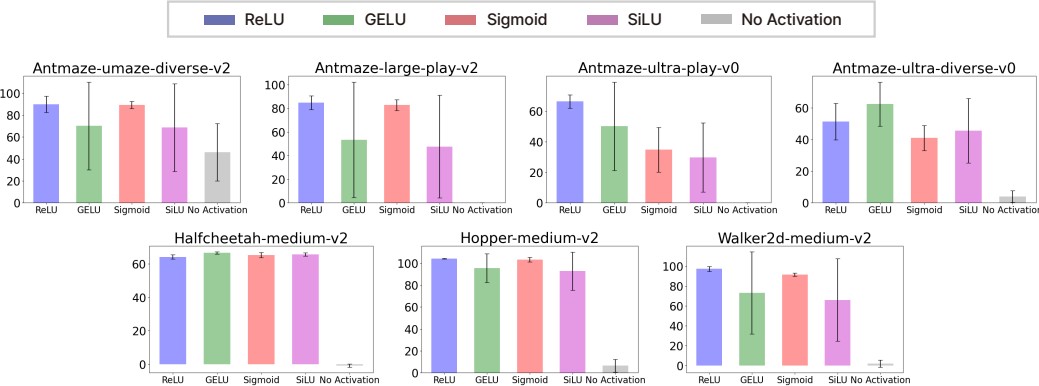

Figure 18: Final normalized score of PARS for offline training averaged over five random seeds with varying activation functions. The error bars represent the standard deviation.

interesting topic for future research.

## H. Computation Cost

We compared the training time and GPU memory usage of PARS with various offline baselines. The comparison was conducted using a single L40S GPU, and the training time was measured over 5000 gradient steps. The baselines were implemented using either PyTorch or JAX. Given that JAX is generally recognized for its speed advantage over PyTorch due to optimizations like just-in-time compilation and efficient hardware utilization (Bradbury et al., 2018), we distinguished the training time and GPU memory usage for PyTorch and JAX with yellow and blue bars, respectively. PARS, implemented in JAX, is indicated by a red bar.

Showing the comparison results presented in Figure 19, PARS has faster training time compared to methods like SAC-N (PyTorch) and MSG (JAX), which use a large number of critic ensembles. Additionally, while SAC-RND and ReBRAC have faster training times than PARS, they use significantly more GPU memory. In contrast, PARS efficiently reduces computation costs by using both less training time and less GPU memory.

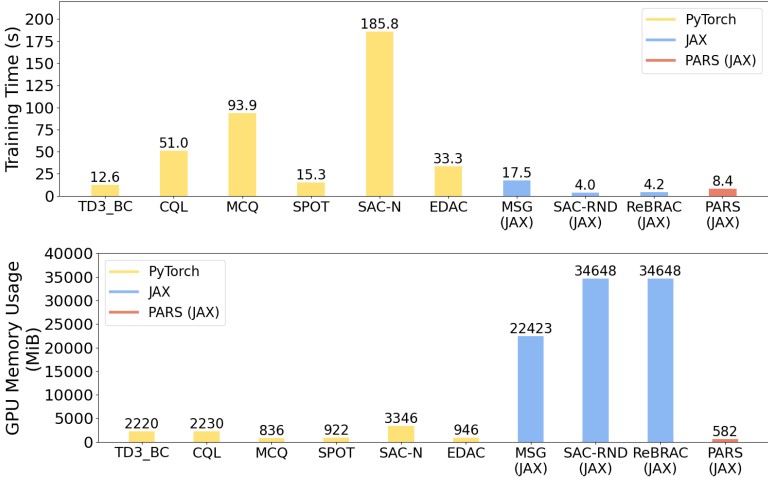

Figure 19: Comparison of PARS's training time and GPU memory usage with various offline baselines. Yellow bars represent PyTorch implementations, blue bars represent JAX implementations, and the red bar represents PARS implemented in JAX.

