# OpenReview forum: "Penalizing Infeasible Actions and Reward Scaling in Reinforcement Learning with Offline Data"
_ICML.cc/2025/Conference — ICML 2025 spotlightposter_

### Official Review · Reviewer_mnXg · 2025-03-14

**Overall Recommendation:** 3

**Summary:**

The paper addresses Q-value extrapolation errors in offline reinforcement learning (RL). It identifies linear extrapolation beyond the data range as a key issue and proposes two methods to mitigate it: (1) reward scaling with layer normalization (RS-LN) and (2) penalizing infeasible actions (PA). These components are integrated into a new algorithm, PARS, which is evaluated on the offline RL benchmark, showing state-of-the-art performance in offline and online fine-tuning, especially in the challenging AntMaze Ultra task.

**Claims And Evidence:**

The paper claims that linear extrapolation in Q-functions leads to overestimation and that RS-LN and PA can effectively mitigate this. Empirical results on the D4RL benchmark support these claims, showing that PARS consistently outperforms baselines.

However, the evidence could be strengthened by additional ablation studies isolating the effects of RS-LN and PA.

**Essential References Not Discussed:**

The paper does not discuss recent advancements in distributional RL, such as IQN or QR-DQN, which could offer alternative solutions to Q-value extrapolation. Additionally, references to prior works on offline-to-online RL transitions, such as model-based fine-tuning strategies, would be benificical.

**Experimental Designs Or Analyses:**

The experiments are extensive and cover a broad range of RL tasks.
But it would be beneficial to explore whether PARS maintains its advantages when applied to more complex tasks beyond D4RL.

**Methods And Evaluation Criteria:**

The methodology is well-grounded, leveraging standard RL benchmarks and evaluation metrics.

However, the choice of hyperparameters for PA and RS-LN could be better justified. Additionally, the comparison to other offline-to-online RL methods would benefit from a discussion on computational efficiency.

**Other Comments Or Suggestions:**

Clarify how the infeasible action penalty interacts with different action space dimensionalities.

**Other Strengths And Weaknesses:**

Strengths:

* A well-motivated approach to mitigating Q-function extrapolation errors.

* Strong empirical performance across diverse RL tasks.

* Simple yet effective implementation with minimal computational overhead.

Weaknesses:

* Lack of ablation study of RS-LN and PA’s effectiveness.

* Limited discussion on computational complexity.

* No exploration of PARS’s applicability to other offline RL benchmarks or real-world tasks.

**Questions For Authors:**

* How does the computational cost of PARS compare to other offline RL methods in terms of training time and memory usage?

* Can RS-LN and PA be effectively combined with model-based offline RL approaches?

* How does PARS perform on tasks with highly stochastic dynamics, where infeasible actions are less clearly defined?

* Could the proposed approach be extended to address policy extrapolation errors, in addition to Q-function extrapolation?

* How robust is PARS to changes in the reward scale, especially when dealing with environments that contain highly varying reward magnitudes or sparse returns?

**Relation To Broader Scientific Literature:**

The paper effectively situates its contributions within the literature on offline RL and critic regularization.

**Theoretical Claims:**

The theoretical analysis of Q-function extrapolation is sound, particularly the discussion of layer normalization’s role in bounding Q-values.

---

> ### Author Rebuttal · Authors · 2025-04-01
>
> We sincerely appreciate the careful review of our work, the clarification of various components of PARS, and your thoughtful suggestions.
> ### [R1] Ablation study separating the effects of RS-LN and PA
>
> Beyond the ablation in Figure 9, we isolated PA in a separate experiment. As shown in **Fig B of the link (https://sites.google.com/view/pars-icml25).**
>
> - None (without LN or PA): Training becomes unstable as the reward scale increases.
> - PA: Helps mitigate extrapolation error, though its effect diminishes at larger reward scales.
> - LN: Performance improves with increasing reward scale.
> - LN and PA: Combining LN with PA further enhances performance by reinforcing downward Q extrapolation.
>
> &nbsp;
>
> ### [R2] Hyperparameter justification
>
> As the reviewer mentioned, $\alpha$ in Eq. (3), the infeasible action distance, and the reward scale are important factors in PARS. The infeasible action distance is analyzed in Figure 11, reward scale in Figure 9, and $\alpha$ in our response to Reviewer QMiG [R3].
>
> &nbsp;
>
> ### [R3] Generalization beyond D4RL
> Thank you for the helpful suggestion. Additionally, **we evaluated PARS on four environments from the NeoRL-2 [Ref.4] benchmark, which includes more complex and realistic problems**, covering a variety of real-world challenges such as delay, external factors, limited data, and safety constraints. As shown in **Table D of the link (https://sites.google.com/view/pars-icml25)**, PARS surpasses previous SOTA even under more realistic benchmark.
>
> &nbsp;
>
> ### [R4] Action space dimensionality and PA behavior
> As shown in Eq.(1), an action is considered infeasible in dimension $i$ if it falls outside predefined thresholds. Thus, $\mathcal{A}_I$ is defined as the union of these infeasible regions across all dimensions. From this union, we sample the same number of infeasible actions as dataset actions per gradient update and assign them a penalty of $Q\_{\text{min}}$.
>
> &nbsp;
>
> ### [R5] Computational efficiency of PARS
> In Appendix H, we compared the computational cost of various offline RL algorithms, showing that PARS is more efficient in training time and GPU memory usage.
>
> &nbsp;
>
> ### [R6] Comparison with uncertainty-based approach
> Thanks for the insightful point. Both distributional and model-based RL incorporate forms of uncertainty—distributional RL models the distribution over Q-values, while model-based RL estimates uncertainty in learned environment dynamics. These uncertainties help with exploration and OOD detection, albeit often at the cost of added complexity and computation. PARS offers a simple and efficient solution built on TD3+BC, without complex uncertainty modeling, yet still achieves strong performance.
>
> We see promise in combining PARS with uncertainty-based methods. In offline RL, penalties may need to differ between interpolation and extrapolation regions, as the latter, lacking clear bounds, can lead to more severe value errors. Penalizing OOD while encouraging downward extrapolation could improve robustness. Similarly, uncertainty-guided exploration during fine-tuning, as reviewers suggested, may boost performance. We'll incorporate these insights in the revision.
>
> &nbsp;
>
> ### [R7] Applicability to stochastic environments
>
> Even in stochastic settings, actions are generally confined within a certain min and max range. Thus, the span defined by these bounds can be regarded as the feasible region. Moreover, instead of sampling infeasible actions that lie just outside the feasible action set, we sample from a sufficiently distant region, using a guard interval. This allows PARS to function without precise knowledge of the feasible-infeasible boundary, supporting robust performance under stochastic dynamics.
>
> &nbsp;
>
> ### [R8] Extension to policy extrapolation
>
> We're not entirely sure what the reviewer means by policy extrapolation, but we interpret it as the policy incorrectly predicting actions for given states. Since the policy is trained via the Q-function, reducing Q-function errors may help mitigate policy errors. However, as Q-function extrapolation and OOD action selection are distinct, directly applying PARS to improve policy accuracy is not straightforward. If we have misunderstood, we would be grateful if the reviewer could kindly clarify.
>
> &nbsp;
>
> ### [R9] Effectiveness of PARS under sparse or varying reward scales
>
> Among our evaluated domains, AntMaze meets the reviewer’s criteria with its sparse reward setting. As shown in Figure 9, performance improves with larger reward scales, highlighting a clear advantage over other baselines. We also analyzed the case where noise is added to the reward; for details, please refer to our response to reviewer snRz [R4].
>
> &nbsp;
>
> [Ref.4] Gao, Songyi, et al. "NeoRL-2: Near Real-World Benchmarks for Offline Reinforcement Learning with Extended Realistic Scenarios." arXiv preprint (2025).
>
> &nbsp;
>
> Thank you once again for your thoughtful review. We hope that our responses have thoroughly addressed your concerns.

---

### Official Review · Reviewer_QMiG · 2025-03-15

**Overall Recommendation:** 4

**Summary:**

In this paper authors address Q-value overestimation problem in offline RL when in the presence of infeasible actions. Authors propose to use two diffferent strategies together, to scale reward and penalize infeasible actions. Layer normalization, as proposed in previous research, is also found to be important to use with reward scaling.

## update after rebuttal

I appreciate authors rebuttal, but taking into account rebuttal and other reviewers opinions I have decided to raise my score from 3 to 4.

**Claims And Evidence:**

Claims are clear, but as the claim is to show that Q-values are not overestimated after the proposed tricks, I do not see any extensive numerical evaluations where we could see what happens to Q-values with and without the proposed tricks. Final benchmarking results are of course shown, but it would be good to tie these things together more strongly.

**Essential References Not Discussed:**

-

**Experimental Designs Or Analyses:**

Analyses provided in numerous Figures are instructive and well designed.

**Methods And Evaluation Criteria:**

Standard bencmarking datasets and environments are used and evaluation criteria as employed in those is applied. Only thing, as mentioned in the previous box is that the estimated Q-values specifically are not shown.

**Other Comments Or Suggestions:**

- Many Figures in Section 6 do not have y-axis labels. Please mark them, sometimes it is not self evident whether it is a reward that is recorded.

**Other Strengths And Weaknesses:**

Paper is interesting and well written. It proposes a novel but simple tricks to make offline RL work in OOD case. Empirical resutls are also very strong.

I have two main questions:
1. infeasible set of actions as in Eqs. 1-2 work only for continuous actions. What happens with discrete actions? In more complex games number of discere actions can happen quite rarely. Leading to similar OOD case as in continuous actions.
2. \alpha paramerer in Eq. 2 seems to be an important one. But I do not see any experimental, nor theoretical analysis on how to set it.

**Questions For Authors:**

-

**Relation To Broader Scientific Literature:**

-

**Theoretical Claims:**

none

---

> ### Author Rebuttal · Authors · 2025-04-01
>
> We sincerely appreciate your positive feedback and your constructive suggestions that allowed us to strengthen areas we may have initially missed.
> ### [R1] Empirical evidence on Q-value overestimation reduction
>
> Thanks for the suggestion. Measuring extrapolation error in high-dimensional state-action spaces is challenging, as it requires sampling outside the dataset’s convex hull—a non-trivial task in such high-dimensional settings. Therefore, following the approach in [Ref.3], we visualized the Q-values learned by applying LN, RS-LN, and RS-LN&PA on the Inverted Double Pendulum with a 1D action space. The visualizations can be found at **link (https://sites.google.com/view/pars-icml25). As shown in Fig A of the link**, and consistent with the discussion in Figure 5 of the manuscript, we observe that training the Q-function without LN leads to divergence. Applying LN prevents this divergence. Furthermore, when reward scaling and PA are also applied, the extrapolation curve bends downward while maintaining a similar overall shape within in-sample actions. We also observe that with reward scaling and PA, the average magnitude of Q-values decreases, suggesting a general reduction in overestimation.
>
> &nbsp;
> ### [R2] Applicability to discrete action spaces
>
> In our work, we focused on the problem of Q-function extrapolation error in continuous action spaces. In discrete action spaces, the action set is finite, making the notion of being “outside” the feasible set less clear—and upward extrapolation of the Q-function less relevant. Since many practical tasks, such as robot control, involve continuous control, and many offline RL algorithms target this setting, we also concentrate on continuous control. We believe that achieving strong performance in this setting, compared to other offline RL algorithms that also focus on continuous control, is a substantial contribution.
>
> &nbsp;
>
> ### [R3] Sensitivity analysis of $\alpha$ hyperparameter
>
> As the reviewer pointed out, when applying PA, the hyperparameter $\alpha$ in eq.(3) is important. It should be set such that assigning a $Q_\text{min}$ penalty to infeasible actions does not interfere with the TD update for actions in the dataset. Accordingly, we set $\alpha$ to a small value in the range of 0.0001 to 0.001. We additionally conducted a sensitivity analysis on $\alpha$. As shown in **Table C of the link (https://sites.google.com/view/pars-icml25)**, PARS performs well when $\alpha$ is in the range of 0.001 to 0.1. On the other hand, if $\alpha$ is too small (e.g., 0.0001), the penalty does not take effect properly, and if it is too large, it starts to interfere with the TD update on the dataset, leading to a decrease in performance.
>
> &nbsp;
>
> ### [R4] Missing y-axis labels in figures
>
> Sorry for the confusion, and thank you for the helpful comment. We'll make sure to clearly specify the y-axis in the revised version.
>
> &nbsp;
>
> [Ref.3] Kim, et al. "Adaptive q-aid for conditional supervised learning in offline reinforcement learning." NeurIPS 2024.
>
> &nbsp;
>
> Once again, we sincerely thank you and hope that our responses have adequately addressed all of your concerns.

---

### Official Review · Reviewer_snRz · 2025-03-15

**Overall Recommendation:** 4

**Summary:**

The paper introduces a method for applying an OOD  penalty using reward scaling and LN. When combined with a modified TD error loss—incorporating a PA penalty—the proposed approach enhances TD3-BC’s performance, achieving strong results on benchmarks, particularly in maze tasks.

I quite like the paper, as it provides a new perspective by investigating how Q-function evaluation interacts with reward tuning and network design. The insights are intriguing and convincing. However, I noticed that many interesting discussions are deferred to the appendix. I suggest moving some of these discussions—such as the reasoning behind using TD3-BC instead of IQL—into the main text for better accessibility.

**Claims And Evidence:**

Most of the claims are supported by explanations and toy tasks. However, I believe the paper would benefit from additional theoretical analysis. Currently, the entire work appears too heuristic, as it lacks theoretical proofs or formally defined mathematical assumptions. Some claims, such as the necessity of a guard interval, are explained but not theoretically justified.

I would be eager to see more theoretical insights if the authors could elaborate on the following points:

1. Why do reward scaling, LN, and ReLU activation lead to good performance, while other combinations do not? Is there any theoretical justification for this choice?
2. Why is a guard interval necessary, and how should its size be determined?
3. With the modified TD loss and the designed $Q_{\min}$, does this modification influence the convergence behavior of Q-value estimation?

**Essential References Not Discussed:**

None.

**Experimental Designs Or Analyses:**

I did not find any explicit issues with the experiments or toy examples.

**Methods And Evaluation Criteria:**

The paper evaluates its method on well-accepted benchmarks.

**Other Comments Or Suggestions:**

None.

**Other Strengths And Weaknesses:**

The paper is novel and easy to follow. The visualizations are well-designed and provide valuable support in understanding the concepts presented. The analysis of interactions between reward scaling, LN, and ReLU offers an intriguing and fresh perspective.

For weaknesses, please refer to my previous section's content and the questions outlined below.

**Questions For Authors:**

1. The definition of OOD relies on the convex hull, but why is it defined this way? I assume the definition is inspired by action interpolation, but is this approach too restrictive? Would it be possible to consider a more flexible, nonlinear boundary for defining OOD actions?

2. I have some concerns regarding reward scaling. In MuJoCo, the rewards are well-designed and provide clean signals. However, in most practical cases, rewards tend to be noisy, and scaling them up to a large magnitude (e.g., $10^4$) could induce **high variance**, potentially leading to training instability or even collapse. Do the authors have any insights on this issue?

3. I find that Figure 1 may not be very informative. Could the authors consider replacing it or, alternatively, moving some discussions from the appendix into the main text? For instance, the discussion on IQL was particularly interesting to me.

4. Q-ensemble is commonly used to mitigate Q overestimation, but is it redundant when using the PA penalty? While I see that the ensemble improves performance, does this imply that the PA penalty does not fully mitigate the overestimation issue? Do the authors have any insights into this?

5. The success of PARS seems to depend on choosing TD3-BC, as it still allows sampling of OOD actions. However, if the policy has high expressive power and can accurately capture all in-sample actions—such as in the case of diffusion policies—would PARS still be effective? If not, does this mean that PARS is mainly beneficial for "less expressive" policies and may not help much for "highly expressive" policies?

I am happy to raise my score if the authors can address my concerns.

**Relation To Broader Scientific Literature:**

None.

**Theoretical Claims:**

No theoretical results are provided in this paper.

---

> ### Author Rebuttal · Authors · 2025-04-01
>
> We sincerely appreciate the positive feedback on our work and the opportunity to clarify any remaining uncertainties.
> ### [R1] Theoretical justification of PARS
> Thank you for the suggestions. As theoretical analysis of deep neural networks with nonlinearities is highly challenging, we empirically validated the effectiveness of RS-LN and leave theoretical aspects for future work. We will clarify this point in the limitations. As noted in Appendix B, we used ReLU due to its popularity and its frequent use in prior works, and exploring other activations better suited to RS-LN, especially with PA, is a promising future direction.
>
> We can show the convergence of PARS, and we will add this to the paper. For this we define the following sets:
> - $\mathcal{ID} = \{(s, a) \mid \beta(a \mid s) > 0\}$
> - $\mathrm{OOD}_{\text{in}} = \mathrm{ConvexHull}(\mathcal{ID}) \setminus \mathcal{ID}$
> - $\mathrm{OOD}_{\text{out}} = \left( \mathrm{ConvexHull}(\mathcal{ID}) \right)^c$
>
> We define the operator $T^\pi_{\mathrm{pars}}$ as:
>
> $
> T^\pi\_{\mathrm{pars}} Q(s,a) =
> \begin{cases}
> T^\pi Q(s,a), & \text{if } (s,a) \in \mathcal{ID}, \\\\
> \mathbb{E}\_{(s',a') \in \mathrm{kNN}(s,a; \mathcal{ID})}[T^\pi Q(s',a')], & \text{if } (s,a) \in \mathrm{OOD}\_{\text{in}}, \\\\
> Q\_{\min}, & \text{if } (s,a) \in \mathrm{OOD}_{\text{out}},
> \end{cases}
> $
>
> where $\mathrm{kNN}(s,a; \mathcal{ID})$ denotes the set of the k-nearest neighbors within $\mathcal{ID}$.
> We show that $T^\pi_{\mathrm{pars}}$ is a contraction under the$\| \cdot \|_\infty$ norm:
>
> - Case 1:$
>   \left|T^\pi_{\mathrm{pars}} Q_1 - T^\pi_{\mathrm{pars}} Q_2\right| = \left|T^\pi Q_1 - T^\pi Q_2\right| \le \gamma \| Q_1 - Q_2 \|_\infty$
>
> - Case 2:$\bigl|T^\pi\_{\mathrm{pars}} Q_1 - T^\pi_{\mathrm{pars}} Q_2\bigr| = \left| \mathbb{E}\_{\mathrm{kNN}}[T^\pi Q_1] - \mathbb{E}\_{\mathrm{kNN}}[T^\pi Q_2] \right| \le \gamma | Q_1 - Q_2 |_\infty$
>
>
> - Case 3:$\left| Q_{\min} - Q_{\min} \right| = 0 \le \gamma \| Q_1 - Q_2 \|_\infty$
>
> Thus, $T^\pi\_{\mathrm{pars}}$ is a contraction, and the Q-function converges accordingly.
>
> In Case 2, computing $\mathrm{kNN}(s,a; \mathcal{ID})$ is costly. Since the Q-function is approximated by a neural network, we let it implicitly learn this behavior instead of adding it to the loss. For the same reason, assigning $Q_{\min}$ can unintentionally affect nearby values. To prevent this, we introduce a **“guard interval”** that excludes actions just outside $\mathcal{A}_F$ from being evaluated as $Q\_{\min}$. In practice, a guard interval of about 1000 (see Figure 11) is sufficient to avoid influencing Q-updates.
>
> &nbsp;
>
> ### [R3] Regarding convex hull definition
> PARS aims to be a simple, efficient algorithm. While nonlinear boundaries are certainly possible, we opt for a convex hull to maintain simplicity.
>
> &nbsp;
>
> ### [R4] Effectiveness of reward scaling in noisy environments
> Assuming the reward is added by noise, scaling the noisy reward increases both true reward and noise proportionally, leaving the signal-to-noise ratio unchanged. To investigate further, we referred to [Ref.2], which analyzes offline RL under reward noise. Following this, we examined how PARS performs in such noisy reward settings. As shown in **Table A of the link** (https://sites.google.com/view/pars-icml25), in the presence of reward noise **the trend of improved performance with increased reward scale still holds. So the core idea of PARS remains valid.**
>
> &nbsp;
>
> ### [R5] Regarding presentation
> In the revision, we will move Appendix D to the main body, possibly by shortening Fig. 1 and others.
>
> &nbsp;
>
> ### [R6] Role of Q-ensemble under the presence of PA penalty
> As noted in Appendix B, PARS primarily targets extrapolation errors outside the convex hull and does not explicitly address OOD actions within it. These actions may still face approximation errors, which could be mitigated through ensemble-based uncertainty estimation. However, as shown in **Table B (https://sites.google.com/view/pars-icml25)**, PARS outperforms SOTA in AntMaze even without an ensemble.
>
> &nbsp;
>
> ### [R7] Dependency on policy expressiveness
> As the reviewer noted, policy expressiveness is important, and methods like Diffusion-QL use diffusion models to enhance it. But, despite using diffusion, Diffusion-QL underperforms compared to PARS and suffers from the high inference cost of diffusion models.
>
> Since policies are derived from the Q-function, we argue that improving Q-network expressiveness is even more crucial. Our work focuses on this, and notably, PARS achieves strong performance with just a simple MLP policy. Ultimately, we believe that combining efforts to improve both policy and Q-network expressiveness is key to building better offline RL algorithms.
>
> &nbsp;
>
> [Ref.2] Yang, Rui, et al. "Towards Robust Offline Reinforcement Learning under Diverse Data Corruption." ICLR 2024.
>
> &nbsp;
>
> Thank you again for the careful review. We hope that our responses have sufficiently addressed the concerns.

---

> > ### Comment · Reviewer_snRz · 2025-04-05
> >
> > Thank you for the authors' detailed response. Most of my concerns have been addressed. However, I still have one remaining question regarding policy expressiveness.
> >
> > From your analysis with IQL, it seems that your method may contradict the use of highly expressive policies. If the policy is already expressive, it is less likely to encounter OOD actions. However, your method appears to benefit significantly from the presence of OOD actions — suggesting that the ability to correct or account for these OOD actions is key to its strong performance.
> >
> > Does this imply that the advantages of your method diminish when using highly expressive policies? In other words, would we see less improvement if the base policy already performs well?

---

> > > ### Author Response · Authors · 2025-04-06
> > >
> > > Thanks for the further response.
> > >
> > > &nbsp;
> > >
> > > Note that high policy expressiveness means the policy's capability to express something complicated more accurately with high fidelity. Consider the following  two major policy extraction methods in offline RL:
> > >
> > > (1) Weighted behavioral cloning (IQL)
> > > - $\max_{\pi}\mathbb{E}\_{s,a \sim D} \left[ e^{\alpha (Q(s,a) - V(s))} \log \pi(a \mid s) \right], ~~~eq.*$
> > >
> > > (2) Behavior-constrained policy gradient (TD3+BC, diffusion-QL)
> > >
> > > - $\max_{\pi} \mathbb{E}\_{s \sim \mathcal{D},\ a' \sim \pi(\cdot|s)} \left[ Q(s, a') \right] - w \cdot \mathbb{E}\_{s \sim \mathcal{D}} \left[ \mathcal{R}(\pi(\cdot|s), \pi_\beta(\cdot|s)) \right],~~~ eq.**$
> > >
> > > where $\mathbb{E}_{s \sim \mathcal{D}} \left[ \mathcal{R}(\pi(\cdot|s), \pi\beta(\cdot|s)) \right]$ is a behavior regularization term.
> > >
> > > So, a policy with higher expressiveness can make the above objective larger by better fitting the policy $\pi(a|s)$ to the desired target function, e.g., $\exp( \alpha (Q(s,\cdot)-V(s))$ in  eq.* in (1).
> > >
> > > &nbsp;
> > >
> > > Now consider the example of Fig C (a) at the following anonymous link (https://sites.google.com/view/pars-icml25-2) and consider the full in-sample policy learning using eq.*  of IQL for this example . As shown in the figure, there are two non-contiguous in-sample regions, with an intermediate $\mathcal{A}\_{OOD-in}$ region. Assuming a perfectly learned Q-function and a highly expressive policy $\pi_\theta$ initialized as $\pi_\theta(a|s) \approx 0,~\forall a$, with random zero-mean weght $\theta$,  solving eq.* using only in-sample data will lead to the learned policy shown in Fig C (b) of the same link. In this case, the policy accurately follows the Q-function within the in-sample regions but assigns near-zero probability elsewhere due to its high capacity and zero initialization. This policy will not generate OOD actions as the reviewer mentioned. But, note that the optimal action for the true Q in Fig C (a) is not the best in-sample action produced by this policy. Therefore, prior work ([Ref. 5]) highlights that **allowing a certain degree of deviation from the in-sample region without straying too far, and broadening the range of considered actions can improve the use of the Q-function over a wider coverage and enhance overall performance when the Q-function is well trained. In addition, when considering online finetuning, assigning near-zero probability to all OOD actions can severely limit online exploration, restricting performance improvements.**
> > >
> > > Note that in the above case of in-sample Q learning, Q function estimation with a more expressive Q function approximator may not increase the performance significantly, as the reviewer noted, because the action is confined to the in-sample region. But, even in this case, there was a nontrivial performance increase of IQL with RS-LN applied to IQL Q-function learning, as shown in Table 5 in Appendix D, repeated in Table E in the link (https://sites.google.com/view/pars-icml25-2). This may be because a more expressive Q-function approximator provides better estimates even for in-sample Q-values, as hinted by the fact that adding PA on top of RS-LN does not improve performance and IQL policy mostly samples in-sample actions.
> > >
> > > &nbsp;
> > >
> > > To overcome the limitation of in-sample policy learning of IQL, we need to go beyond in-sample policy learning.  One way is to consider the policy learning, eq. ** in (2). e.g., TD3-BC in which $\mathcal{R}(\pi(\cdot|s), \pi_\beta(\cdot|s))= \left\| \pi(s) - a \right\|^2$.  Here, the policy is allowed to generate OOD action $a'$ from $s'$ as shown in eq. ** but not too far from in-sample actions.  Then, the Q-values of OOD actions, especially those in $\mathcal{A}_{OOD-out}$, matter now, and suppressing the upward trend in  $\mathcal{A}\_{OOD-out}$ becomes crucial. Here, RS-LN and PA play an important role, creating strong synergy with (2), which leads to strong performance in both offline training and online finetuning.
> > >
> > > &nbsp;
> > >
> > > **In summary, applying RS-LN to Q function approximator yields performance gain both in IQL and TD3-BC, but the major influencing mechanisms are different (i.e., more accurate in-sample domain Q learning for IQL and more accurate in-sample domain Q learning + slight increase of action range + suppression of upward trend at $\mathcal{A}_{OOD-out}$ for TD3-BC), and the gain is more significant for TD3-BC.**
> > >
> > > &nbsp;
> > >
> > > Another point is that applying eq.* in (1) requires direct computation of $\log \pi(a \mid s)$. However, diffusion models do not provide an explicit probability distribution. Thus, it is not easy to use diffusion models for the in-sample policy learning, eq.* in (1). Indeed, Diffusion-QL [Ref.6], uses eq.** in (2).
> > >
> > >
> > > &nbsp;
> > > Thank you.
> > >
> > >
> > > &nbsp;
> > >
> > > [Ref.5] S. Park et al. "Is Value Learning Really the Main Bottleneck in Offline RL?." NeurIPS 2024
> > >
> > > [Ref.6] Z. Wang,  J. Hunt, and M. Zhou. "Diffusion Policies as an Expressive Policy Class for Offline Reinforcement Learning." ICLR 2023

---

### Official Review · Reviewer_rMGd · 2025-03-16

**Overall Recommendation:** 4

**Summary:**

The manuscript introduces penalizing infeasible actions and reward scaling (PARS), a method for discouraging value overestimation caused by extrapolation in offline reinforcement learning (RL). The proposed method uses reward scaling and layer normalization, which are shown to work together to increase the feature resolution. This has the effect of reducing the similarity between gradient updates corresponding to actions found in the dataset and actions that lie outside the convex hull of “feasible actions”. In addition, the authors also propose to penalize out-of-distribution actions that lie far outside the region of feasible actions. Extensive benchmarking experiments demonstrate that the method consistently achieves superior performance compared to many baselines. Analytical experiments provide additional insights on the contribution of each component and the effects of hyperparameters.

## update after rebuttal
The authors' detailed rebuttal provided important clarifications and additional experimental results which support the efficacy of the proposed method. I am now more confident in recommending the paper for acceptance and have accordingly raised my score to a 4.

**Claims And Evidence:**

* PARS successfully addresses the value overestimation problem caused by extrapolation in offline RL. This is well supported by better performance in experiments on many environments, including the challenging AntMaze Ultra and Adroit relocate-cloned
* Penalizing infeasible actions during online fine-tuning is effective for stabilizing training. Shown in experiments on two difficult environments (Figure 10)

**Essential References Not Discussed:**

I think the following work is related:
* Gulcehre, C., Colmenarejo, S. G., Wang, Z., Sygnowski, J., Paine, T., Zolna, K., ... & de Freitas, N. (2021). Regularized behavior value estimation. arXiv preprint arXiv:2103.09575.

**Experimental Designs Or Analyses:**

My main concern is whether the comparison with other algorithms is fair in the sense that several hyperparameters of PARS were tuned per environment. Is that the case for every score that you report from other papers and did you have the same search budget for hyperparameters of other methods that you ran yourself?

**Methods And Evaluation Criteria:**

The proposed method is mostly well-motivated. The benchmarks show clear performance improvements over competing methods. The authors also present several ablation studies and analytical experiments investigating the effect of hyperparameters such as the reward scale or the distance of infeasible actions from the feasible action region.

**Other Comments Or Suggestions:**

* I think it’d be worthwhile looking at how PARS affects policy churn (Schaul et al., 2022)
* Can you comment on whether PARS has the potential to improve safe RL in the sense that it reduces sampling of actions that are far from the explored action space?

Schaul, T., Barreto, A., Quan, J., & Ostrovski, G. (2022). The phenomenon of policy churn. Advances in Neural Information Processing Systems, 35, 2537-2549.

**Other Strengths And Weaknesses:**

Strength: Extensive analytical experiments that show efficacy of each contribution or support design choices such as the distance of sampled infeasible actions from the feasible action region (Figure 11)

Weakness: I think the motivation for penalizing infeasible actions is not very strong. I understand that it probably helps to intentionally push down the Q-values at far regions of the action space, but I’d like to know your thoughts on why it improves the performance compared to only using reward scaling + layer normalization.

**Questions For Authors:**

* Is layer norm only applied to the pen-ultimate layer, i.e. the one before the output weights, or to every layer?
* In Figure 10 (left), the LN online fine-tuning seems to reduce performance. Why is that?

**Relation To Broader Scientific Literature:**

The proposed method is novel and addresses the important problem of value overestimation due to extrapolation error in offline RL that has been studied before. It is the first to make a distinction of OOD actions that are within the convex hull of the actions in the dataset and those outside. Discouraging high Q-values for actions outside this region of “feasible” actions seems to be very effective at mitigating extrapolation errors.

**Theoretical Claims:**

N/A

---

> ### Author Rebuttal · Authors · 2025-04-01
>
> We sincerely appreciate your positive evaluation of our contribution and suggestions regarding various prior studies and potential directions for extending PARS.
> ### [R1] Comparison of tuning search budget for hyperparameters with baselines
>
> As noted in our manuscript, we tuned around 8 hyperparameter configurations, varying $\alpha$, TD3+BC $\beta$, and reward scale. For context, prior offline RL works we compare against also performed environment-specific tuning: ReBRAC used 25 configurations for MuJoCo and Adroit and 192 configurations (grid over 4 actor betas, 4 critic betas, 4 actor learning rates, and 3 critic learning rates) for AntMaze, SAC-RND used 19 (MuJoCo) and 9 (AntMaze), and SPOT, MSG, and MCQ used 6, 12, and 7, respectively. SAC-N and EDAC used 7 and 12.
>
> As shown above, most of the prior works **we compare against adopt a similar level (some even more extensive) of hyperparameter tuning, which supports the fairness of our comparisons.** For TD3+BC, IQL, and CQL, we report results from their original papers (1–2 configurations), and ReBRAC [Ref.1] provides extensively tuned results for them—yet PARS still performs strongly.
>
> Importantly, as shown in Appendix G, PARS achieves strong AntMaze performance even with a **single** hyperparameter configuration, outperforming other baselines.
>
> &nbsp;
> ### [R2] Motivation for PA
>
> As discussed in Section 3.2, our goal is to encourage the learned Q-function to exhibit downward extrapolation for OOD actions that lie outside the convex hull of the dataset. To achieve this, RS-LN first increases the expressivity of the neural network, which naturally enables downward extrapolation by preventing gradient updates that would otherwise increase the Q-value positively for OOD actions. In addition, PA makes this process more explicit by penalizing infeasible actions, thereby enforcing downward extrapolation. As demonstrated in the ablation study in Figure 9, RS-LN has a significant effect on its own, but applying PA on top further enforces downward extrapolation, contributing stability and performance of offline RL training.
>
> &nbsp;
> ### [R3] Relation to prior work
>
> Thank you for pointing out the relevant prior work. We will include the study in the revision. In particular, R-BVE also aims to mitigate Q-function extrapolation errors using SARSA-based in-sample Q-learning and ranking loss, performing policy improvement only once at the end. However, this single-step approach may achieve insufficient performance improvement in complex environments or where rewards are sparse. While the ranking loss aligns Q-values based on high-reward trajectories, it risks performance degradation when applied to low-reward data, requiring additional mechanisms like soft filtering that increase implementation complexity.
>
> In contrast, PARS directly addresses Q-value extrapolation, especially outside the convex hull of the data, by combining RS-LN and PA. This provides a stable foundation for repeated policy improvement and achieves both simplicity and superior performance.
>
> &nbsp;
> ### [R4] Regarding policy churn
>
> Thanks for the insightful suggestion. The referenced paper studies policy churn—sudden shifts common in value-based methods like DQN. In contrast, PARS builds on TD3+BC with a behavior cloning term in the actor, which helps suppress such changes. While this likely reduces policy churn, continual critic updates may still cause some churn, potentially aiding exploration. Quantitatively analyzing churn in critic-regularized methods like PARS would be a valuable direction for future work.
>
> &nbsp;
> ### [R5] Implications for safe RL
>
> In our study, we define infeasible actions as those lying outside the set of feasible actions. However, this definition can be extended by incorporating various constraints depending on the objectives of different tasks. In particular, by establishing clear criteria for unsafe infeasible actions and designing the system to penalize such actions accordingly, the framework can be naturally extended to safe RL.
>
> &nbsp;
> ### [R6] Where layer normalization is applied
>
> We applied LN after all linear layers, including the input layer, except for the final output layer.
>
> &nbsp;
> ### [R7] Regarding figure 10 (left)
>
> As the reviewer mentioned in another comment, a key issue during online fine-tuning is that changes in the distribution can lead to unstable learning. Since PA was applied during offline training, we observed that removing PA during online fine-tuning led to instability. On the other hand, maintaining PA throughout resulted in more stable learning and led to significant performance improvements.
>
> &nbsp;
>
> [Ref.1] Tarasov, Denis, et al. "Revisiting the minimalist approach to offline reinforcement learning." NeurIPS 2023.
>
> &nbsp;
>
> We sincerely appreciate your thoughtful review once again. We hope that our responses have thoroughly addressed all of your comments.

---

> > ### Comment · Reviewer_rMGd · 2025-04-03
> >
> > I thank the authors for their detailed response including clarifications regarding hyperparameter tuning for baselines, the relation to Gulcehre et al., and the placement of layer norm modules. I have read the detailed rebuttals for the other reviews of this manuscript, which also include additional analytical experiments that support the efficacy of the proposed method.
> > I will increase the score to a 4.
> >
> > Minor feedback:
> > Regarding R7, I think it'd make sense to just add a very short note indicating that the fine-tuning experiments are *after pre-training with PA*, even though that appears somewhat obvious in hindsight.

---

> > > ### Author Response · Authors · 2025-04-03
> > >
> > > We sincerely thank the reviewer for continuously engaging with our response and for adjusting the recommendation. Your valuable suggestions and comments have helped improve our work. We will also incorporate the additional comment regarding R7 into the revision.

---

### Decision · Program_Chairs · 2025-05-01

**Decision:**

Accept (spotlight poster)

**Comment:**

This work introduces PARS, an algorithm that aims to bridge the gap between offline reinforcement learning and online finetuning. Concretely, the authors introduce a method that is a strong offline algorithm which is also amenable to online finetuning, and show that this approach greatly outperforms prior work in a large number of domains with minimal hyperparameters. The work achieves this by building on existing work using layer norm that penalizes OOD actions (e.g., RLPD), adding reward scaling and explicit action penalization in infeasible regions.

The reviewers were very positive on this work, and highlighted its very strong empirical results and interesting and novel insights into value learning in RL more broadly. The authors also had a strong rebuttal, which helped eliminate concerns that the reviewers had around experiments and theoretical insights.

As a result, this is clearly a strong work that pushes the SoTA on an important topic on the RL community, and therefore should be accepted into the program.